# The relationship between extra-tropical cyclone intensity and precipitation in idealised current and future climates

Victoria A. Sinclair[1] and Jennifer L. Catto[2]

[1]Institute for Atmospheric and Earth System Research / Physics, Faculty of Science, University of Helsinki, PO BOX 64, FI-00014
[2]Faculty of Environment, Science and Economy, University of Exeter, Exeter, United Kingdom

**Correspondence:** Victoria Sinclair (victoria.sinclair@helsinki.fi)

**Abstract.** Extra-tropical cyclones (ETCs) are the main cause of precipitation in the mid-latitudes and there is substantial evidence that ETC-related precipitation will increase in the future. However, little is known about how this will impact on the dynamical strength of ETCs, and whether the impact will differ for different types of ETCs. We quantify the linear relationship between maximum vorticity and ETC related precipitation in the current and idealised future climates, and determine how this relationship depends on the structure and characteristics of the ETC. Three 10-year long aqua-planet simulations are performed with a state-of-the-art global model, OpenIFS, that differ in their specified SST distributions. A control simulation, a uniform warming simulation, and a polar amplification simulation are performed. ETCs are objectively identified using the feature tracking software TRACK and k-means clustering is applied to the ETC precipitation field to group the ETCs into clusters with similar precipitation structures. In all experiments, ETCs with stronger maximum vorticity are associated with more precipitation.

For all cyclones considered together, we find that the slope of the linear relationship between maximum cyclone vorticity and ETC precipitation is larger in the uniform warming and polar amplification simulations than in the control simulation. We hypothesise that if an increase in precipitation in warmer climates were to feed back, via diabatic heating and potential vorticity anomalies, onto the dynamical intensity of the ETCs, precipitation and vorticity would increase at similar rates and hence the slope of the linear regression line between precipitation and vorticity would remain similar. Our results indicate that there is either no feedback or that the increase in vorticity due to diabatic heating is masked by the decrease in the Eady growth rate which occurs in both the the uniform warming and polar amplification simulations compared to the control.

The k-means clustering identifies four distinct and physically realistic types of ETCs which are present in all experiments meaning that the average precipitation patterns associated with ETCs are unlikely to change in the future. The strongest dependency between ETC maximum vorticity and precipitation occurs for ETCs that have most precipitation associated with the warm front. ETCs with the heaviest precipitation along the cold front, which are the most intense storms in terms of maximum vorticity, also exhibit a strong dependency between precipitation and maximum vorticity but this dependency is weaker and has a smaller correlation coefficient than the warm front ETCs. Not all ETC types exhibit a strong dependency between precipitation and maximum vorticity. ETCs located at high latitudes with weak precipitation show little dependency due to the lack of moisture, whereas ETCs with the precipitation located mainly in the centre of the ETCs have the weakest linear regression

slope which is likely due to the lack of upper-level forcing. These results stress that despite small changes in the strength of the cyclones, the precipitation increases are large, indicating potential future increases in flooding associated with cyclones.

## 1 Introduction

Extra-tropical cyclones (ETC) constitute a large part of the Earth's circulation, transporting energy and momentum poleward. These weather systems are also the dominant cause of day-to-day weather in the mid-latitudes. Hawcroft et al. (2012) showed that the majority of precipitation (70 - 80%) in the mid-latitude storm track regions is associated with ETCs and Pfahl and Wernli (2012) found that ETCs are responsible for a high percentage of precipitation extremes. Some, but not all, ETCs can have extremely large amounts of precipitation associated with them which can lead to flooding and thus to significant societal and economic impact. Therefore, it is important to understand what controls the amount of precipitation related to ETCs and how ETC related precipitation may change in the future.

ETCs have been extensively studied over the past century starting from Bjerknes (1919) and Bjerknes and Solberg (1922) who developed the well-know Norwegian model, a conceptual model of the structure of ETCs. Already from these early studies it has been known that precipitation develops in ETCs in regions where air is ascending, which typically occurs along the frontal zones. Climatological studies of the occurrence of fronts have found that in the main storm track regions, up to 80% of the total precipitation can be associated with fronts (Catto et al., 2012), and an even larger proportion of extreme precipitation events (Catto and Pfahl, 2013). These exact proportions depend on the front identification and dataset used, but indicate the importance of these features.

The warm conveyor belt (WCB), a coherent ascending air stream which originates in the boundary layer of the warm sector of ETCs, was first described by Browning (1971), Harrold (1973) and Carlson (1980) and is also known to be a source of precipitation in ETCs. Using a climatology of WCBs developed based on the ERA-Interim reanalysis dataset (Madonna et al., 2014), Pfahl et al. (2014) showed that 70–80% of precipitation extremes in some regions are associated with the WCB airstream. This feature, when associated with fronts, also gives a higher chance of an extreme precipitation event (Catto et al., 2015). Ascent in the warm conveyor belt is forced by warm air-advection but is also enhanced by diabatic heating. This diabatic heating can also influence the cyclone dynamics via the production of a low-level positive potential vorticity (PV) anomaly, which develops below the localised maximum in diabatic heating (Stoelinga, 1996). This positive PV anomaly can potentially feed back onto the intensity of the ETC by enhancing the low-level circulation. Binder et al. (2016) used a classification method to show that the WCB flow can contribute to rapid intensification of extratropical cyclones, but only in situations where there is strong enough upper-level forcing.

ETC intensity and the associated precipitation are strongly linked. We expect to see more precipitation in strong ETCs where there is strong forcing for ascent (Milrad et al., 2010; Dai and Nie, 2020). Field and Wood (2007) combined satellite observations with mean sea level pressure (MSLP) from reanalysis to understand how ETC precipitation and cloudiness depend on ETC strength and moisture availability. They found that precipitation increases with both ETC strength (as quantified by the mean surface wind speed within a 2000 km radius) and moisture, but that the deepest cyclones in terms of MSLP

do not produce the most precipitation despite having the strongest winds. Pfahl and Sprenger (2016), using ERA-Interim reanalysis data, also found a strong relationship between ETC strength (in terms of wind speeds) and precipitation but also noted that this relationship varied strongly with latitude, with a very weak relationship at high latitudes, due to limitations in moisture availability. They also showed that the relationship between precipitation and intensity was strongest when looking at the precipitation from 12 hours prior to the maximum intensity. This result, that maximum precipitation occurs before the maximum intensity of the ETC, is supported by Booth et al. (2018) who found that the maximum vorticity of an ETC occurred after the precipitation maximum in most ETCs (>70%), while only 20% of ETCs had their precipitation maximum after the vorticity maximum. Owen et al. (2021) focused on extremes (including both precipitation and winds and their co-occurrence) and found that ETCs associated with the extremes tend to have higher intensity than those that are not associated with extremes.

Associated with the increased moisture due to the higher temperatures projected in the future, precipitation intensity (especially that of extremes) is expected to increase (Allen and Ingram, 2002). Many studies have shown this is also the case for ETC-related precipitation. For example, Zhang and Colle (2018) analysed ETCs in eastern North America and the western Atlantic in 10 models from phase 5 of the Coupled Model Intercomparison Project (CMIP5) and found that ETC-related precipitation increases by up to 30% between the historical and future climate simulations. Michaelis et al. (2017), in regional pseudo global warming experiments, also found ETC precipitation increases in a warmer climate while Hawcroft et al. (2018) showed that the number of ETCs associated with extreme precipitation may triple by the end of the century. Kodama et al. (2019) performed two high resolution global climate model simulations; one current climate and a future climate simulation, and found that precipitation associated with intense ETCs increases at 7% / 1 K warming, whereas when all ETCs are considered, precipitation increases only at 3% / 1K warming. Similar results were also found by Reboita et al. (2021) for Southern Hemisphere ETCs when regional climate model simulations were analysed. Yettella and Kay (2017) analysed a 30-member initial condition climate model ensemble and found that most of the increase in the intensity of precipitation associated with ETCs resulted from the thermodynamic effect of increased temperature, rather than a dynamical effect. Studies have also suggested that as well as an increase in ETC precipitation intensity in the future, the size of the cyclones and the area of the precipitation may increase (Reboita et al., 2021; Dai and Nie, 2022)

Climate change is expected to impact on the intensity of ETCs themselves. One reason for this is the predicted changes to the large-scale atmospheric state, and in particular, changes to baroclinicity. Arctic amplification reduces the low-level temperature gradient and thus baroclinicity whereas tropical upper troposphere warming acts to increase the upper-level temperature gradient and baroclinicity. In addition, changes to the vertical temperature profile (stability) can also impact baroclinicity. Furthermore, in a warmer climate, forcing from latent heat release is projected to increase and potentially may increase the intensity of ETCs alongside the increase in precipitation (Sinclair et al., 2020; Binder et al., 2022). However, the number of ETCs of a given intensity (i.e. the frequency distribution of ETC intensity) is not projected to change much (Priestley and Catto, 2022) even in high-emissions scenarios, indicating that either the diabatic feedback is weak or that the combination of factors listed above impact the number of ETCs in different direction. In contrast, the extreme ETCs are projected to increase in intensity in terms of vorticity, maximum wind speeds, and the footprint of high wind speeds (Pfahl et al., 2015; Sinclair

et al., 2020; Priestley and Catto, 2022; Dolores-Tesillos et al., 2022), suggesting that not all types of ETCs respond in the same manner to climate change.

Idealised models have been used extensively in the past to understand cyclone dynamics (Simmons and Hoskins, 1978; Thorncroft et al., 1993) and, more recently, to understand ETC dynamics, precipitation and intensity in a warming climate. Two idealised modelling approaches have been used to study how ETCs response to a warmer climate, the first is baroclinic life cycle experiments (e.g. Boutle et al., 2011; Kirshbaum et al., 2018; Rantanen et al., 2019) and the second is aqua-planet simulations (e.g. Kodama et al., 2014; Pfahl et al., 2015; Sinclair et al., 2020). Both modelling approaches offer advantages over fully complex models, being able to better identify physical mechanisms for changes in ETCs. In this study we employ an aqua-planet model configuration, and include full physics, as previously used in Sinclair et al. (2020). This setup allows for a full range of ETCs to develop (e.g. Catto, 2016), while removing complexities associated with land-sea contrasts of temperature, moisture, and surface drag.

It is clear that ETC-related precipitation is projected to increase in intensity in the future. A question remains as to whether this will impact the dynamical strength of the cyclones. In other words, it is unclear how the relationship between precipitation and intensity might change in the future. Furthermore, there are many different dynamical structures of extratropical cyclones (e.g. Evans et al., 1994; Sinclair and Revell, 2000; Catto, 2016, 2018; Binder et al., 2022) and how this relationship depends on the type of ETC has yet to be quantified. Booth et al. (2018) used a subsetting method to group similar cyclones according to their precipitable water, and we will take a related approach by applying a clustering method to the precipitation structure to group ETCs. Using idealized aqua-planet simulations of a control climate, a uniform global warming scenario, and an Arctic amplification scenario, we will determine potential future changes in the cyclones and associated precipitation in the different cyclone clusters.

The questions we aim to address in this research are:

1. What is the relationship between precipitation intensity and cyclone intensity in the current and potential future climates?

2. How does the relationship between cyclone precipitation and cyclone intensity depend on the type of cyclone?.

3. How does the variability of precipitation structures associated with extratropical cyclones change in the future climate, and are certain types of cyclones more or less common?

Section 2 describes the numerical model, OpenIFS, that we use and the setup of the simulations that are performed. The methods used to analyse the output from these simulations are described in section 3. In section 4, the basic climatology of the three simulations is described before the relationship between precipitation and cyclone intensity is discussed in section 5. An analysis of the different types of ETCs that occur is presented in section 6 and how the relationship between precipitation and ETC intensity depends on the type of ETC is presented in section 7. Conclusions are presented in section 8.

## 2 Model Simulations

### 2.1 OpenIFS

The numerical simulations are performed with the state-of-the-art global numerical weather prediction model, OpenIFS Cy43r3v1. OpenIFS is a version of the Integrated Forecast System (IFS) used operationally at the European Centre for Medium Range Weather Forecasts (ECMWF). The dynamical core and the physical parameterizations in OpenIFS are identical to those in the same cycle of the full IFS. However, unlike the IFS, OpenIFS does not include the data assimilation package nor is it coupled to an ocean model. OpenIFS is available under licence to academic and research institutions. The equivalent version of the IFS

(Cy43r3) to the version of OpenIFS used in these simulations was operational between July 2017 and June 2018. The complete documentation of IFS Cy43r3 is available online at https://www.ecmwf.int/en/publications/ifs-documentation

### 2.2 OpenIFS simulations

The numerical experiments conducted in this study utilise OpenIFS configured as an aqua-planet in which there is no land and the surface of the Earth is completely covered by ocean. Three simulations are performed and all are initialised from a real

atmospheric state selected at random. However, some modifications are required to make the initial conditions suitable for an aqua-planet simulation. First, the land-sea mask is modified to be zero (ocean) everywhere and the surface geopotential is also set to zero. The atmospheric states are then extrapolated to the new, flat surface. Surface pressure is also adjusted to reflect the removal of surface topography. All simulations have a diurnal cycle in incoming radiation but no annual cycle. During the simulations the incoming solar radiation is fixed at the equinoctial value and is thus symmetric about the Equator. The

simulations are all run for 11 years and the first year of simulation is discarded to account for model spin-up. Output fields are written every 6 hours. The simulations are all run at a horizontal resolution of T255 (approximately 80 km) and have 60 model levels between the surface and the model top at 0.1 hPa.

The three aqua-planet simulations performed only differ from each other in terms of their prescribed sea surface temperature (SST) distributions (Fig. 1). In each experiment, the SST distribution is analytically specified as only a function of latitude and

the SSTs are held constant through the simulation. The simulations represent a control, a case of uniform warming, and a case of polar amplification. The control simulation uses the QObs SST distribution proposed by Neale and Hoskins (2000). The QObs SST profile is a simple analytical function of latitude which has a maximum of 27°C on the equator and reaches 0°C at 60°N and which Neale and Hoskins (2000) state resembles the observed zonal mean SST distributions. The SST distribution in the uniform warming simulation (referred to as SST4) is the QObs distribution but warmed everywhere by 4 K. The SST4

experiment is motivated by previous aqua-planet simulations with the same fixed SST forcing that have been performed as part of CMIP5 (aqua4K, Taylor et al. (2012)) and CMIP6 (amip-p4k, Eyring et al. (2016)). The polar amplification simulation uses the QObs SST distribution between 45°S and 45°N and polewards of these latitudes, the SSTs are set to 5 °C. This choice is made to test the impact of warming the high latitudes without altering the tropics, and hence this experiment is not intended to represent reality. Rantanen et al. (2022) show, using reanalysis data sets, that large parts of the Arctic are warming at a rate of

0.75 K per decade (evaluated between 1979–2021) and locally some regions are warming faster than 1 K per decade. Therefore,

our 5 K increase is consistent with the amount of warming expected over a 50 - 60 year period and thus is not unreasonably large.

Similar aqua-planet experiments with OpenIFS have previously been presented by Sinclair et al. (2020). A few minor differences exist between the simulations presented here and those previous simulations. Firstly, the version of OpenIFS differs (Cy43r3 versus Cy 40r1), secondly, the horizontal resolution differs (T255 versus T159) and lastly, Sinclair et al. (2020) did not modify the surface pressure to account for the removal of topography from the randomly selected real initial conditions and therefore have lower climatological values of surface pressure than in this study.

## 3 Analysis methods

### 3.1 Cyclone Tracking

In all experiments, extra-tropical cyclones (ETCs) are objectively identified and tracked using TRACK (Hodges, 1994, 1995). ETCs are identified as localised maxima in the 850-hPa relative vorticity field truncated to T42 spectral resolution. Wave numbers smaller than wave number 5 are also set to zero to remove planetary scale features. Six-hourly input data is used to identify the ETC tracks. Initially, all ETCs in the Northern Hemisphere are identified, however to ensure that only synoptic-scale and mobile ETCs are retained for analysis we only retain ETC tracks that last for at least 2 days and travel 1000 km. To remove weak ETCs, tracks with a maximum T42 850-hPa relative vorticity of less than $1\times10^{-5}\mathrm{s}^{-1}$ are also excluded from the analysis. Similar filtering has been applied in many other studies that employ TRACK (e.g. Dacre and Gray, 2009; Hodges et al., 2011; Priestley et al., 2020). In addition to these standard filters, we also require that ETCs reach their maximum vorticity at a latitude north of 30°N and exist 24 hours before the time of maximum intensity.

### 3.2 Cyclone composites

Following the same method as Catto et al. (2010) and Dacre et al. (2012), composites of a range of meteorological variables are computed for the ETCs at different offset times relative to the time of maximum vorticity (t=0 hr). Negative offset times indicate that the composite is valid before the time of maximum intensity and hence when the ETC is intensifying. In contrast, positive offset times mean that the composite is valid during the decaying part of the ETC lifecycle. The first step in creating the composites is to select which ETCs to include in the composite. Previous studies have often select the strongest 50 - 200 ETCs (e.g. Catto et al., 2010; Flaounas et al., 2015; Vessey et al., 2022) or have selected the subset of ETCs to be composited by considering their geographic location. Here we create composites of different types of ETCs based on their precipitation patterns as identified by k-means clustering (more details in section 3.3). The second step in creating the composites is to regrid the meteorological field from the regular latitude-longitude grid that the OpenIFS output is on, to a spherical grid centred on the cyclone centre (the location of the maximum T42 vorticity obtained from TRACK). The spherical grid has a radius of 18 degrees and consists of 40 grid points in the radial direction and 360 grid points in the angular direction. After the meteorological fields have been interpolated onto this grid, these fields are then rotated so that all ETCs are travelling to the

east. Finally to obtain the ETC composite, the meteorological values on the radial grid at each offset time are averaged. Thus, the composite extra-tropical cyclone is the simple arithmetic mean of the selected ETCs.

Compared to previous studies, the composites analysed here are produced by averaging a much larger number of ETCs (e.g. > 2000). This may result in a large degree of smoothing if there is large variability between the ETCs in the selected population. Therefore, we investigated how sensitive the results of the composite mean precipitation are to the number of ETCs included in each composite by creating clusters with sub-samples of ETCs. Overall, the main conclusions are not strongly sensitive to the number of ETCs included in composites (not shown).

### 3.3 Cyclone Clustering

K-means clustering (Lloyd, 1982) is used to separate the ETCs into different groups with different precipitation structures. As input to the k-means clustering algorithm, we use the regridded precipitation field (convective plus large-scale precipitation) within a 12 degree radius of the ETC centre 12 hours before the time of maximum vorticity. Twelve degrees was selected as the radius as, based on plotting many individual ETCs, this provided a good balance between ensuring that all precipitation clearly related to the ETC was included and that precipitation related to another nearby ETC was excluded. It is also consistent with the radius used to attribute precipitation to ETCs in Hawcroft et al. (2012). However, to determine how sensitive the results are to this choice of radius, the clustering was repeated but with various different radii (4.5, 8, 12 and 18 degrees) of the ETC centre and these results are discussed in the Supplementary material and shown in Figure S1.

As we want to allocate each ETC to a cluster based on their spatial distribution of precipitation, and not on the absolute precipitation values, we use quantile mapping (scikit-learn's quantile transform function, Pedregosa et al. 2011) to normalise the precipitation field of each individual ETC to a uniform distribution. In practice, this means that for each ETC, each grid cell on the polar grid is given a value between 0 and 1 and all values (for each individual ETC) represent a uniform distribution. Quantile mapping is robust to outliers and also performs well with sparse and semi-sparse arrays. Although we do not consider the absolute amount of precipitation when clustering the ETCs, in our subsequent analysis we do compute and analyse the ETC total precipitation (see section 3.4).

A disadvantage of k-means clustering is that the number of clusters is not automatically selected by the algorithm and instead must be specified in advance by the user. Furthermore, it most situations there is no clear, ideal number of clusters. Usually, the optimal number of clusters is determined by trying a number of different options and calculating various measures that quantify how similar an element (i.e one ETC in our case) is to its own cluster compared to other clusters. We tested 2 to 19 clusters and for each clustering computed the silhouette score (Rousseeuw, 1987). The silhouette score ranges from -1 to +1 where large positive values indicate that the element is very well matched to its cluster. In contrast, low or negative values mean that an element is poorly matched or potentially mis-classified. The silhouette score is computed for each element (each ETC) and then the final silhouette score is the average of all elements.

Figure S2 shows the silhouette score for all three experiments as a function of cluster number. Based on these results, we use 4 clusters. A smaller number of clusters (e.g. 2) gives higher scores but little meaningful information can be obtained from only two clusters. We also select k=4 as to enable us to more easily compare the three experiments we wanted to have

the same number of clusters in each experiment and for k=4, all experiments still have moderate silhouette scores and the AA experiment even exhibits a localised maximum. For k=5, and especially k=6, both the control and AA experiments see a reduction in the silhouette score although this is not the case for the SST4 simulation. In addition, we also computed the Euclidean distance between each individual ETC and the centroids of all clusters to ensure that ETCs are, on average, assigned to the most appropriate cluster and that the clusters are distinct from each other. Additional details are provided in section 3 of the supporting material.

### 3.4  Precipitation diagnostic

To enable comparison between thousands of ETCs in each experiment, a succinct diagnostic for the ETC precipitation is required. For each ETC, we compute the average precipitation rate, $P_{ave}$, which is defined as

$$P_{ave} = \sum_{i=1}^{m} P_i \frac{A_i}{A_T} \tag{1}$$

where $P_i$ is the precipitation rate in each grid cell $i$ and $m$ is the number of grid cells within a given radius of the cyclone centre on the spherical grid and where $P_i$ exceeds 1 mm / 6 hr. $A_T$ is the total area covered by $m$ grid cells and $A_i$ is the area of each individual grid cell $i$. Hence, we average the precipitation rate but only over grid cells where precipitation is actually occurring.

### 4  Climatology of the three experiments

The global mean 2-m temperature and precipitation, averaged over the 10 years of simulation, for each simulation is presented in Table 1. The control simulation has a global mean 2-m temperature of 286.7 K (13.5°C), which is similar to that of the real Earth (global mean surface temperature of 288 K, Hartmann (2015)), and a global mean precipitation rate of 3.18 mm day$^{-1}$ which is 20% larger than the real Earth (average precipitation of 2.66 mm day$^{-1}$, Hartmann (2015)) due to the absence of land in these simulations. The SST4 simulation is 4.1 K warmer than the control simulation and also has a larger global mean precipitation rate (3.63 mm day$^{-1}$) than the control simulation. This precipitation increase equates to an increase of 3.5% per degree of warming which is slightly larger than the most likely range of 2 - 3% found in recent climate model simulations (Douville et al., 2021). The AA simulation is 1.1 K warmer than the control simulation and when global precipitation is considered, there is a very slight decrease (0.01 mm day$^{-1}$) compared to the control simulation. However, when only the temperature and precipitation in the mid-latitudes (25 - 70°N) are considered, the AA simulation warms by 1.6 K and the precipitation increases by 0.06 mm day$^{-1}$, equivalent to an increase of 1.54% per degree of warming.

The zonal and time mean atmospheric state in the control experiment and how it changes in the SST4 and AA experiments are shown in Fig. 2. In the control simulation, the zonal wind has a maximum of 53 m s$^{-1}$ located at 31.2°N and 175 hPa (Fig. 2a) and the Eady growth rate (Eady, 1949), a measure of baroclinicty, (Figs. 2c) has a maximum value of 0.93 d$^{-1}$. Uniform warming (SST4) causes the jet to move polewards and upwards (Fig. 2a), indicating a lifted tropopause, and causes a decrease in the Eady growth in the mid-to-upper tropopshere on the equatorward side of the jet and poleward of 70°N/S. The decrease

in the Eady growth rate is caused by an increase in tropospheric stability in these regions (Fig. S5c), rather than a decrease in the meridional temperature gradient, which barely changes (Figs. S5a). Near the tropopause, there is a vertically orientated dipole in how the Eady growth rate responds to uniform warming which is due to the tropopause height increasing and thus the stability changing. Polar warming in the AA simulations causes the jet to move equatorwards (Fig. 2b). The response of the Eady growth rate to polar warming is a decrease in the low-to-mid troposphere on the poleward side of the jet and an increase in the mid-to-upper troposphere at high latitudes. The low-level decrease is due to a decrease in the meridional potential temperature gradient (Fig. S5b) whereas the increase at high latitudes is related to a decrease in stability in the polar atmosphere (Fig. S5d). Thus, based on the changes to the zonal and time mean Eady growth rate, weaker or fewer ETCs could be expected in the SST4 simulation whereas a more complex picture emerges in the AA simulation. Fewer or weaker ETCs could be anticipated in the main storm track region but more, or stronger, ETCs may be able to develop at high latitudes.

Basic statistics of the characteristics of the objectively identified ETCs are shown in Table 2. There are 578 fewer ETCs identified in the SST4 simulation compared to the control simulation, however the decrease is not statistically significant at the 95% confidence level when a t-test is performed on the yearly totals (N=10) assuming unequal variance. In contrast, 440 more ETCs are identified in the AA simulation than in the control and this difference is statistically significant at the 95% level. Notably, there is more year-to-year variation in the number of ETCs in the SST4 simulation compared to both the control and AA simulations (Table 2).

The mean and median values of the maximum relative vorticity of the ETCs in the control and SST4 simulations are very similar (Table 2). The full distributions (Fig. 3a) are also very similar and do not differ statistically when a two-sided t-test is performed, despite the reduction in the Eady growth rate. However, the SST4 simulation has a broader distribution with both more weak and more strong ETCs. In the SST4 simulation 12.1% of ETCs have a maximum vorticity exceeding $10 \times 10^{-5} s^{-1}$ whereas only 10.3% of ETCs exceed this threshold in the control simulation. The mean and median maximum relative vorticity of all ETCs in the AA simulation is smaller than in the control simulation (Table 2) which is also evident in the full distribution (Fig. 3a). A one-sided t-test shows that the maximum vorticity values in the AA experiment are statistically significantly weaker than in the control simulation. This is consistent with the fewer very strong ETCs in the AA simulation compared to both the control and SST4 simulations and with the reduction in the Eady growth rate in the AA simulation compared to the control (Fig. 2).

The genesis and lysis latitudes of the ETCs in each simulation are also considered and in all simulations exhibit roughly Gaussian distributions. The genesis and lysis latitudes of ETCs move polewards in SST4 compared to the control whereas the opposite behaviour occurs in the AA simulation (Table 2 and Fig. 3b,d). These changes are consistent with the latitudinal changes in the jet stream position (Fig. 2). Furthermore, the same response is found when the median latitude of maximum vorticity is considered (Table 2).

Lastly, as the focus on this study is on ETC related precipitation, we also considered the distributions of the ETC averaged precipitation in each simulation (Fig. 3c). In all experiments, there is a localised peak for very weak (1.25 - 1.5 mm / 6 hr) precipitation amounts. This is partly because of the 1 mm / 6 hr threshold used in the precipitation diagnostic (Equation 1). This localised peak is more pronounced in the AA experiment as there are more ETCs with weak precipitation compared

to the other two experiments. The control simulation has more ETCs with weak to moderate precipitation ( 2 - 3 mm / 6 hr) than either the SST4 or AA simulations, which we hypothesise may be due to the control simulation being the coldest simulation. The SST4 simulation has many more ETCs with precipitation amounts exceeding 4 mm / 6 hr than in either the control or AA simulations, which show very similar distributions for heavier precipitation amounts. A t-test confirms that there is no significant difference at the 95% level between the control and AA simulation in terms of ETC related precipitation. In contrast, t-tests show that ETCs in the SST4 experiment have statistically significantly more precipitation associated with them than ETCs in the control simulation at the 95% level.

The distribution of the maximum vorticity of ETCs differs between the three simulations. Therefore, an Analysis of Covariance (ANCOVA) statistical test is applied here to determine if the ETC related precipitation differs in the different experiments, after controlling for a covariate, which in this case is the maximum vorticity of the ETC. This allows us to test the null hypothesis, which is that ETCs with the same intensity (measured in terms of the relative vorticity) have the same amount of precipitation associated with them in each experiment. When the control and SST4 simulations are compared using ANCOVA, the returned p-value is $1.11 \times 10^{-186}$ which is less than 0.05 meaning that we can reject the null hypothesis. Hence, the ANCOVA analysis shows that ETCs in the SST4 simulation have more precipitation than those in the control simulation even after accounting for the differences in ETC maximum vorticity. This is as expected, given that there is no significant difference between the maximum vorticity distributions yet notable differences in the precipitation. Of more interest is to apply ANCOVA to the control and AA simulations, as the AA simulation has weaker cyclones yet the same amount of ETC related precipitation. After controlling for the differences in ETC maximum vorticity, we find that the ETCs in the AA simulation have statistically significantly (p-value of $2.21 \times 10^{-26}$) more precipitation associated with them: i.e., an ETC in the AA experiment will have more precipitation than an ETC in the control simulation for the same maximum vorticity.

## 5 The relationship between precipitation and cyclone intensity

We hypothesise that if precipitation increases in warmer climates, and if it were to feed back, via diabatic heating and potential vorticity anomalies, onto the dynamical intensity of the ETCs, precipitation and vorticity would increase at similar rates. Hence, if this occurs and is the dominant process acting, then the slope of the linear regression line between precipitation and vorticity would remain similar in all experiments. To test this hypothesis the slope of the linear regression and the Pearson's correlation coefficient between the maximum vorticity and ETC-related precipitation at different offset times relative to the maximum vorticity were considered (Table 3). In all experiments, large correlations and positive slopes occur between maximum intensity and precipitation 24, 12 and 0 hours before the time of maximum intensity. The smallest correlations and weakest slopes occur between maximum vorticity and precipitation 24 hours after the time of maximum vorticity. Here we focus on the relationship between maximum vorticity and precipitation 12 hours before the time of maximum vorticity, as shown in Figure 4, as this is when the largest slopes and correlation coefficients occur. The same as Figure 4, but for precipitation 24 and 0 hours before the time of maximum vorticity, are shown in the Supplementary material (Figs. S6 and S7).

Both the SST4 and the AA simulations have steeper linear regression slopes and thus a stronger dependency between maximum vorticity and precipitation than in the control simulation. To determine if these slopes are statistically different, boot-strapping is applied. Bootstrapping quantifies how much random variation in the slope and intercept values of the fitted linear model are due to small changes in the input data. For each experiment, we re-sample the data 5000 times using a sample size equal to that of each original data set. Thus, some pairs of data are represented multiple times in any one individual bootstrap sample while other pairs are not selected at all. For each re-sample, a linear regression is made and the slope and intercept of this model calculated. For the control simulation the estimated slopes vary from 0.261 - 0.287 mm (6 hr)$^{-1}$ / $10^{-5}$ s$^{-1}$, for the SST4 simulation the values are 0.313 - 0.348 mm (6 hr)$^{-1}$ / $10^{-5}$ s$^{-1}$, and for the AA simulation the estimated slopes range from 0.298 - 0.326 mm (6 hr)$^{-1}$ / $10^{-5}$ s$^{-1}$. The resulting distributions of the computed slopes (Figure S8 in Supplementary material) for each experiment are then compared using a student's t-test and we find that the slopes are all statistically significantly different at the 99% level. Thus, we can conclude that there is a greater dependency between precipitation and maximum vorticity in the SST4 experiment than in both the control or the AA experiment, and, that the AA experiment has a greater dependency between precipitation and maximum vorticity than the control. This means that for the same increase in maximum vorticity, precipitation increases more in the SST4 and AA experiments compared to the control and hence that our hypothesis that both precipitation and vorticity would increase at similar rates is not supported by the simulation results.

Since the SST4 and AA experiments are both warmer than the control experiment, the increase in precipitation could be explained by the Clausius-Clapeyron relationship between temperature and vapour pressure. However, the same percentage increase in precipitation would be expected for ETCs of all intensities, which is not the case. One potential explanation is that the increase in precipitation is larger for ETCs with stronger dynamical forcing and ascent as they are better able to convert the additional moisture into precipitation, which would also explain the increase in slope. However, the increase in slope could also be interpreted as there being no (or weak) feedback onto the vorticity of the ETC from the enhanced precipitation via a diabatically produced low-level PV anomaly as, if this was the case, similar slopes would be found in all experiments. Finally, the increased slope in SST4 and AA could result from an increase in precipitation and may still indicate an increased diabatic feedback on vorticity. However, this diabatically-driven increase in maximum vorticity might be masked by the counteracting decrease in the Eady growth rate in both simulations relative to the control, similar to what was found by Büeler and Pfahl (2019). These arguments will be further explored in section 7 as it is plausible that different explanations are valid for different types of ETC.

The most spread and thus smallest Pearson's correlation coefficient (0.574) occurs in the SST4 simulation suggesting that in this simulation, the dynamical intensity of the cyclone has less control on the amount of precipitation than in the other simulations. The largest correlation coefficient (0.635) and hence least spread occurs in the AA simulation. However, in all three simulations the correlation coefficients are not exceptionally large and we hypothesise that a large degree of the spread is caused by different types of ETCs having different relationships between their intensity and their precipitation.

## 6 Different types of ETCs according to their precipitation patterns

Figures 5a-d show the composite mean total precipitation for each of the four clusters identified by the k-means clustering in the control simulation 12 hours before the time of maximum intensity (additional times are shown in Fig. S9). The precipitation pattern differs between all four mean ETCs meaning that the k-means clustering has successfully separated the ETCs into different classes.

### 6.1 ETC composites in the control simulation

The composite mean ETC in Fig. 5a has a large area of very heavy precipitation, exceeding 7 mm / 6 hr. The position of the warm and cold fronts, and the warm sector, is evident in the 850-hPa potential temperature (Fig. 6a) and total column water vapour (TCWV, Fig. 7a). The precipitation is heaviest near where the warm and cold fronts meet but also extends along the cold front and parts of the warm sector. A strong (maximum value 1.1 PVU), localised low-level PV centre is evident close to the cyclone centre, but poleward and slightly upstream of the heaviest precipitation (Fig. 6a). The convective precipitation (Fig. 7a) is moderate in intensity and only located on the poleward parts of the cold front. Hence, this composite mean is subsequently referred to as the "cold front" ETC. This cold front ETC has a very narrow warm sector which is immediately equatorward of the ETC centre. Since this ETC is meridionally extended yet zonally confined it exhibits similarities with the Norwegian cyclone model (Bjerknes, 1919). This cold front ETC is also located directly in the left-hand exit region of a strong ($>50$ ms$^{-1}$) jet streak where strong forcing for ascent due to positive vorticity advection can be expected. A strong upper-level PV anomaly visible in the 315K isentropic potential vorticity (IPV, Fig. 8a) is also present immediately upstream of the ETC centre. Consequently, this mean ETC is a strong system with a minimum MSLP of 978 hPa (Fig. 8a). Downstream of the ETC centre, there are low values of IPV on the 315 K surface, which is likely related to the strong ascent in the warm conveyor belt (Fig. 8a) and the diabatic erosion of PV downstream. The 315K IPV pattern also shows this ETC is wrapped up in a cyclonic manner.

The precipitation pattern associated with the second identified ETC is of moderate intensity (up to 3.5 mm / 6 hr) and is mainly associated with the warm front, a bent-back warm front (which hooks around the poleward and upstream side of the ETC centre) and in the warm sector (Fig. 5b). Thus, this composite is here-in-after referred to as the "warm front ETC". The warm front ETC has a broad warm sector, evident in both the 850-hPa potential temperature (Fig. 6b) and TCWV (Fig. 7b), which is shifted well downstream of the ETC centre. A low-level PV maxima is evident with the largest values (0.91 PVU) close to the ETC centre and moderate values extending downstream along the warm front. The strongest ascent (Fig. 8b) is co-located with the heaviest precipitation on the warm front. Very little ($< 1$ mm $(6$ hr$)^{-1}$) convective precipitation is associated with this mean ETC (Fig. 7b). The 850-hPa potential temperature (Fig. 6b) shows that both the warm and cold front have similar temperature gradients. The zonally broad nature of this mean ETC suggests that it resembles the Shapiro-Keyser conceptual model (Shapiro and Keyser, 1990). Similar to the cold front ETC, the warm front ETC is also located on the poleward side of a jet streak. However, this ETC is not directly in the left-hand exit region, but is located 10 degrees (on the rotated polar grid)

poleward of the 45 ms$^{-1}$ jet streak (Fig. 8b), which likely explains the weaker vertical motion and precipitation associated with
385 this ETC. This ETC also has strong IPV anomaly at upper levels and is also shows signs of cyclonic wave breaking.

The precipitation associated with the third type of ETC identified by the k-means clustering is shown in Figure 5c. This ETC
is also located in the left-hand exit region of a jet streak but the jet streak is weaker than those associated with either the warm
front or cold front ETC (Fig. 8c). The precipitation associated with this ETC has a small spatial extent and is mainly focused
on the ETC centre. Therefore, this composite mean is subsequently referred to as the "centre" ETC. Despite the weaker jet
streak, and high minimum MSLP (991 hPa) the centre ETC has heavier precipitation and stronger ascent than the warm front
ETC, yet a slightly weaker (0.85 PVU) low-level PV maximum (Fig. 6c). The increased precipitation is likely explained by
the moister air advected towards the centre of this ETC: the TCWV values reach to 13 g kg$^{-1}$ in the ETC centre in this small
cyclone but only to 10 g kg$^{-1}$ in the warm front ETC. In agreement with the increased moisture, this ETC has more convective
precipitation than the warm front ETC. This ETC also differs from the warm and cold front ETCs in that it does not have a
395 large low pressure centre and only a small-scale closed circulation. Furthermore, at upper levels this ETC has a weaker IPV
anomaly and is less cyclonically wrapped up than the other two ETCs, remaining more of an open wave at upper levels. This
weaker upper level forcing may explain why the MSLP is not very low.

The precipitation pattern of the fourth and last ETC type is shown in Figure 5d. This ETC differs considerably from all other
ETCs. Firstly, it is much weaker in terms of vertical motion (Fig. 8d) than all other ETCs, although a closed circulation is
400 evident with a minimum MSLP of 991 hPa (which is almost the same minimum MSLP as seen in the centre ETC). Secondly,
this weak ETC is not located in the jet exit but rather is located far poleward of the jet stream and does not have a pronounced
upper level IPV anomaly upstream of the ETC centre. Thirdly, there is only a very weak low-level PV maximum co-located
with the ETC centre (Fig. 6d) Lastly, this ETC is located in a cold and dry air mass; the TCWV values near the ETC centre
are around 7 g kg$^{-1}$ (Fig. 7d) and the 850-hPa temperature values are 278K (Fig. 6d). The total and convective precipitation
associated with this ETC are also much weaker than in all other mean ETC. Therefore, this composite mean is subsequently
referred to as the "weak" ETC.

So far, only the mean structure of each cluster has been presented. Figures 9, 10, and 11 show the distributions of the
ETC related precipitation, maximum vorticity and the latitude of the maximum vorticity for each cluster. The shapes of the
precipitation, maximum vorticity and latitude of maximum vorticity distributions are similar for the cold front, warm front and
410 centre ETC whereas the weak ETCs have very different shaped distributions, particularly for the latitude of maximum vorticity.

The distributions for each cluster within the control simulation can be compared to each other using Student's t-test to deter-
mine how different the clusters are and also to explain the differences in the mean ETC structures. When the full distributions
for each cluster in the control simulation are considered, the cold front ETCs (Fig. 9a) have statistically significantly more
precipitation associated with them than the ETCs in the other three clusters (Figs. 9d, g, k), which is in agreement with the
415 mean values presented in Figure 5. The cold front ETCs also have statistically significantly larger values of maximum vorticity
compared to all other clusters (Fig. 10a, d, g, k) and also reach their maximum vorticity values further south (mean latitude of
44.4°N) than ETCs in the other three clusters.

The centre ETCs in the control simulation have statistically significantly more precipitation associated with them than the warm front ETCs despite that the warm front ETCs have stronger maximum vorticity values (the difference is statistically significant). The centre ETCs reach their maximum vorticity at an average latitude of 45.1°N, which is 2.3° farther equatorward than the ETCs in the warm front cluster which reach their maximum at an average latitude of 47.4°N (Fig. 11). This difference is found to be statistically significant when the full distributions are considered, that is, the warm front ETCs reach their maximum vorticity more poleward than the centre ETCs do. The weak ETCs in the control simulation have the smallest maximum vorticities but also occur at much higher latitudes than the other types of ETC.

## 6.2 ETC clusters in the SST4 and AA simulations

Figures 5e-h and i-l show the precipitation patterns and MSLP for the four composite mean ETCs in the SST4 and AA simulations (additional times for the SST and AA experiments are shown in Figure S10 and S11 respectively). The main result is that in all three experiments, very similar types of ETCs in terms of their precipitation patterns occur; the cold-front, warm-front, centre and weak cyclone identified in the control simulations also occur in the SST4 and AA simulations. Furthermore, the dynamical structure and location relative to the jet streak (Fig. 8) is very similar for all clusters in the SST4 and AA experiments compared to the corresponding cluster in the control simulation. However, some differences do exist, particularly in terms of the absolute values of thermodynamic variables.

The mean ETCs in the SST4 experiment all have heavier total precipitation and convective precipitation than in the control simulation although the spatial patterns are very similar. Likewise, the mean composite ETCs in the SST4 experiment all have much higher values of TCWV and 850-hPa potential temperature associated with them. This is consistent with an overall warmer and thus moister environment. Related to this, the SST4 composite means have lower IPV values on the 315K isentrope, caused by this isentrope being lower in the troposphere in the warmer simulation. The minimum MSLP is 1 - 3 hPa lower in the SST4 mean ETCs compared to their related clusters in the control simulation. Interestingly, the vertical motion at 700 hPa is almost identical in the corresponding composites in the control and SST4 simulations, indicating that the large increase in precipitation is not directly related to changes in the ascent. The only discernible difference in the ETC structure occurs for the warm front ETC. In the SST4 simulation this is associated with a weaker jet streak compared to the control (Fig. 8b, f).

Now the distributions of precipitation, maximum vorticity and latitude of maximum vorticity for each cluster in the SST4 simulation are compared to the corresponding cluster in the control simulation. A summary of the statistically significant differences is presented in Figure 12. When the distributions of ETC precipitation for each cluster in the SST4 simulation are compared to the corresponding cluster in the control simulation, one-sided t-tests show that all clusters in the SST4 experiment have statistically significantly more precipitation associated with them (Figs. 9, 12). The cold front ETCs in the SST4 simulation have statistically significantly larger values of maximum vorticity compared to the corresponding cluster in the control simulation (Figs. 10, 12). However, when the warm front and centre ETC clusters are considered, there is no statistically significant difference between the distributions of maximum vorticity between the SST and control simulations. Furthermore, the weak ETC cluster has statistically significantly smaller values of maximum vorticity compared to the corresponding cluster in the control. This means that how the maximum vorticity of ETCs responds to uniform warming depends on the type of ETC.

This result is consistent with the result presented in section 4 - that uniform warming does not change the maximum vorticity of ETCs - but also indicates that looking at all ETCs together can mask notable changes. The cold front, warm front and centre ETCs occur at higher latitudes in the SST4 simulation in comparison to their counterparts in the control simulation (Figs. 11,12). However, in the SST4 experiment, the weak ETCs occur at slightly lower latitudes (mean latitude of 60.8°N) than the weak ETCs in the control simulation.

We now compare the cluster means in the AA simulation to the corresponding clusters in the control simulation. All ETCs have significantly more precipitation associated with them in the AA simulation (Fig. 5). However, the increases are relatively small except for the warm front ETC, which has an increase of almost 1 mm (6 hr)$^{-1}$ in the precipitation rate on the bent-back warm front. In addition, although the MSLP patterns are very similar between the AA and control composite mean ETCs, the minimum MSLP is 2 - 4 hPa higher in the AA simulation than in the control. The jet structure, 700-hPa ascent, and upper level IPV fields are also very similar between the AA ETC cluster means and those in the control.

When the ETC precipitation distributions for each cluster in the AA simulation are compared to the corresponding cluster in the control simulation (Fig. 9), we find that the ETCs in the cold front, warm front and centre ETCs clusters have more precipitation associated with them in the AA simulation than in the control (summary shown in Fig. 12). In contrast, the precipitation associated with the weak ETC cluster has no significant difference between the AA and control simulation. The cold front, centre, and weak ETC clusters have statistically significantly weaker maximum vorticity values in the AA simulation compared to in the control simulation (Figs. 10, 12). The maximum vorticity of the ETCs in the warm front cluster does not differ between the AA and the control simulation. All clusters have their latitude of maximum vorticity more equatorward in the AA simulation compared to the control (Fig. 11). This result is statistically significant for all clusters (Fig. 12) and consistent with the changes in genesis and lysis latitude presented in Figure 3.

### 6.3 Frequency of occurrence of the different ETCs

To help answer the question of whether different types of ETCs are more or less common in the different climates, the number of ETCs in each cluster for each experiment and the relative occurrence of the 4 types of ETCs is shown in Figure 13. In the control simulation the absolute number of ETCs in the warm front, cold front and the centre ETCs is very similar. However, there are slightly more ETCs in the weak cyclone cluster. In the SST4 simulation, again the weak cyclone cluster has the most ETCs. The cold front ETC cluster is the second most common but closely followed by both the warm front and centre ETCs. In the AA simulation there are many more weak ETCs than the other three ETC types. Furthermore, the number of weak ETCs in the AA simulation is much larger than the absolute number and relative occurrence of weak ETCs in the control or SST4 simulation. This indicates that Arctic amplification and the associated increase in high latitude temperatures and decrease in stability (Fig. S5d) leads to many more weak ETCs developing. This is despite the decrease in the meridional temperature gradient which, from an energy transport point of view, would indicate fewer ETCs are required to transport heat polewards. However, these weak ETCs are mainly at high latitudes and likely contribute little to the total poleward heat transport. Furthermore, the increase in the number of weak ETCs does not come at the expensive of stronger ETCs; there are simply more ETCs in the AA experiment but the majority of the extra ETCs are weak.

## 7 Effect of ETC type of the relationship between precipitation and cyclone intensity.

Figure 14 shows the relationship between maximum vorticity and total precipitation 12 hours before the time of maximum intensity for each cluster in each experiment. In the control simulation, the largest correlation coefficient and slope of the linear regression between precipitation and vorticity occur for the warm front ETC (Fig. 14d). A moderate correlation coefficient and slope are also present for the cold front ETC in the control simulation (Fig. 14a) yet there are very small correlations and regression slopes in the centre and weak ETCs (Fig. 14g, j).

The steeper slope in the warm front ETC, compared to the cold front ETC in the control simulation, means that vorticity increases less for the same increase in precipitation in warm front ETCs compared to in cold front ETCs. However, despite the steeper slope in the warm front ETCs, the cold front ETCs produce more precipitation than the warm front ETCs of the same intensity and this is true for all ETC intensities e.g. for weak and strong ETCs. This may be because the warm front ETCs occur at slightly higher latitudes (compare Fig. 11a and d) where there is less moisture available and thus less precipitation occurs for a ETC of the same intensity. Alternatively, as the cold front ETCs have a larger low-level PV anomaly (compare Fig. 6a and b), this may indicate that the cold front ETCs have a stronger diabatic feedback than the warm front ETC and hence vorticity is enhanced more for the same amount of precipitation in the cold front ETC compared to the warm front ETC.

The small linear regression slope and correlation coefficient found for the weak ETCs in the control simulation (Fig. 14j) is very likely explained by the high latitude of these ETCs (Fig. 11j) and the limited amount of moisture available in the locations they develop in and move through. The lack of dependency between vorticity and precipitation in the centre ETCs (Fig. 14g) is also notable as these ETCs occur at low- to mid-latitudes (Fig. 11g) where moisture is not limited. There are ETCs included in the centre ETC cluster which have small values of maximum vorticity yet moderate to large amounts of precipitation associated with them. These centre ETCs are quite convective (Fig 7c) and hence the precipitation amount is not heavily influenced by the large-scale ascent or the strength of the circulation. Furthermore, the centre ETCs have a weaker upper-level PV anomaly (Fig. 8c) compared to the cold (Fig 8a) and warm front (Fig 8b) ETCs in the control simulation.

In the SST experiment, the strongest linear regression and largest correlation coefficient is again found in the warm front ETCs (Fig. 14e). Weak correlations and linear regressions are found in the centre and weak ETCs (Fig. 14h, k). Uniform warming causes the linear regression slopes for all four clusters to increase (compare the first and second column in Fig. 14). Hence, for all clusters precipitation increases more with uniform warming than does maximum vorticity. Our hypothesis was that if increased precipitation in warmer climates feeds back, via diabatic heating and low-level PV anomalies, onto the dynamical intensity of the ETCs, precipitation and vorticity would increase at similar rates and hence the slope of the linear regression line between precipitation and vorticity would remain similar. However, as the slopes for all clusters increase with warming (Fig. 14) this means that there is either a weak diabatic feedback or that the increase in vorticity caused by an increase in diabatic heating is offset by a decrease in vorticity caused by a reduction in the large-scale baroclinicity. The largest relative increase in the linear regression slope in the SST4 experiment compared to the control occurs in the weak ETC as with uniform warming these ETCs are less moisture limited than in the control simulation. The cold front ETC has a larger relative increase

in the slope than the warm front ETCs with warming, indicating that the cold front ETCs may be better at converting the extra

moisture into precipitation than the warm front ETCs.

As was the case in the control and SST4 simulations, in the AA experiment the largest Pearson's correlation coefficient and regression slope occurs for the warm front cyclone cluster (Fig. 14f) and weak correlation coefficients and small linear regression slopes occur for the centre (Fig. 14i) and weak (Fig. 14l) ETCs. Relative to the control simulation, polar amplification causes the slope of the linear regression between maximum vorticity and precipitation to increase in the warm front, centre

and weak ETC clusters but to decrease in the cold front ETC (compare the first and third columns in Fig. 14). The largest relative increase in the regression slope compared to the control simulation occurs for the centre ETC (+62%), however, the linear regression slope still remains very small ($0.104$ mm $(6 \text{ hr})^{-1}$ / $10^{-5}$ s$^{-1}$) in the centre ETCs. A moderate increase in the linear regression slope occurs for the weak ETC (+24%), which is due to the increase in moisture in the high latitude regions, where these ETCs occur, in the AA experiment compared to the control. The moderate increase in the slope for the warm

front ETCs means that precipitation is increasing more than vorticity. This is consistent with Fig. 12 which shows that the maximum vorticity of the warm front ETCs is not statistically different between the control and AA simulations. The lack of an increase in vorticity suggests that diabatic processes do not act to intensify warm front ETCs, or that any increase in vorticity by diabatic processes is counteracted by a decrease in vorticity driven by a reduction in baroclinicity. The decrease in the linear regression slope in the cold front ETC in the AA simulation compared to the control is small (-2.3%) but bootstrapping shows

the difference to be statistically significant. This small decrease occurs as although ETCs of all intensities see an increase in precipitation with polar amplification, ETCs with lower maximum vorticity values that are included in the cold front cluster see a slightly larger increase in precipitation than the strongest ETCs in the same cluster.

## 8 Conclusions

In this study, we investigated the relationship between the maximum intensity of extra-tropical cyclones and their precipitation,

how this may change in the future, and also how it depends on the type of ETC. Three aqua-planet simulations were performed with the state-of-the-art numerical prediction model, OpenIFS, differing only in terms of their prescribed SST distributions. A control simulation, a uniform warming and a polar amplification experiment were conducted.

First the response of the zonal mean large-scale environment was analysed. Uniform warming resulted in a poleward shift in the jet stream, no change to the low-level meridional potential temperature gradient and an increase in the Brunt Väisälä

frequency. Consequently, the Eady growth rate decreased with uniform warming. Polar amplification resulted in an equatorward shift in the jet stream and a narrowing of the baroclinic zone. The Brunt Väisälä frequency did not change with polar warming in the mid-latitudes, which combined with a reduction of the meridional potential temperature gradient, lead to a decrease in the Eady growth rate. In the mid-to-upper troposphere at high latitudes the Brunt Väisälä frequency decreased causing an increase in the Eady growth rate. The impact of uniform warming and polar amplification on the cyclone statistics was then

investigated. Uniform warming resulted in an increase in ETC precipitation and a small poleward shift in the storm track, consistent with the poleward shift in the jet. No statistically significant change to the mean or median maximum vorticity was

detected but more extreme ETCs in terms of their maximum vorticity occurred with uniform warming. Given the decrease in the Eady growth rate with uniform warming, this response of ETC maximum vorticity indicated that diabatic processes may be acting to counteract the decrease in vorticity expected from decreasing baroclinicity, especially for the strongest ETCs.

Polar amplification led to no change in ETC precipitation, weaker ETCs in terms of their maximum vorticity consistent with the decrease in Eady growth rate, and a small equatorward shift of the storm track. These results of how the jet position and ETC number and intensity responded to warming agree with previous results from both idealised modelling experiments and fully coupled realistic climate models (e.g. Yettella and Kay, 2017; Priestley and Catto, 2022) which gives confidence in the relevance and robustness of our idealised simulations.

In all three experiments, there was a positive correlation between the ETC maximum 850-hPa vorticity and average precipitation, which is in agreement with previous studies (e.g. Pfahl and Sprenger, 2016). When all ETCs were considered together, the largest linear regression slope was in the uniform warming (SST4) experiment and was smallest in the control. Thus, the same absolute increase in ETC strength in SST4 corresponded to a larger increase in ETC-related precipitation than in the control simulation. SST4 was the warmest and wettest simulation globally which likely explains the strongest slope, especially

when the results of Yettella and Kay (2017), who showed most of the increase in ETC precipitation is due to thermodynamic aspects, are considered. Theoretically, heavy precipitation causes strong diabatic heating which, in turn, could increase the intensity of the ETC via the production of a low-level positive vorticity anomaly. However, in our warmer experiments, precipitation strongly increased but there was not any large increase in the intensity of ETCs i.e. the slope was larger in SST4 than in the control. This implies that the increased diabatic heating does not feedback strongly onto the intensity of the ETCs

in these experiments or that the diabatic feedback is counteracted by a decrease in large-scale baroclinicity. The horizontal resolution of these simulations was relatively coarse at 80 km, and if similar experiments were performed with a much higher resolution model, the amount of diabatic heating and the strength of the low-level PV anomaly would very likely increase; this was recently shown to happen by Choudhary and Voigt (2022). Potentially, the diabatic feedback may become more visible at high resolution, in which case, we would expect that the slopes of the linear regression lines between maximum vorticity and

precipitation would be similar - albeit shifted to larger values in the warmer simulations. However, whether a feedback occurs may depend on the type of cyclone. In the one case study that Choudhary and Voigt (2022) considered, the stronger diabatic heating and potential vorticity at higher resolution did not result in a lower MSLP as the cyclone was strongly dominated by thermal advection and diabatic heating played a secondary role in its intensification.

The variability (spread) in the relationship between precipitation and ETC intensity also varied between the experiments.

The smallest Pearson's correlation coefficient ($r$) was found in the SST4 experiment suggesting there is larger variability in ETC-related precipitation in the SST4 experiment than in either the control or polar amplification experiment. All experiments did show considerable spread (the correlation coefficients ranged between 0.574 and 0.635) which suggests that not all ETCs have a strong connection between ETC intensity and precipitation. This large spread motivated us to cluster the ETCs into 4 groups using k-means clustering and investigate how the relationship between precipitation intensity and cyclone intensity

depends on cyclone type.

The clustering proved to be successful. Four distinct and physically realistic mean cyclones were identified by compositing all ETCs allocated to each cluster together. Each cluster (mean composite ETC) could be labelled based on their precipitation patterns: a cyclone where most precipitation was associated with the cold front (cold front ETC), a cyclone where most precipitation was associated with the warm front (warm front ETC), a small-scale cyclone with most precipitation located near the cyclone centre (centre ETC) and a small-scale, high latitude cyclone with weak precipitation again focused on the cyclone centre (weak ETC).

The first notable result from the clustering was that the same four mean ETCs were identified in all three experiments despite the clustering being done independently for each simulation. This suggests that neither uniform warming nor polar amplification will lead to any notable changes in the average spatial patterns of precipitation associated with ETCs. The second main result of the clustering was that the absolute number and relative occurrence of the weak ETC, which was found to develop in cold air masses far poleward of the jet stream, increased considerably with polar amplification, despite the large-scale meridional temperature gradient decreasing. We hypothesise that this is because the polar amplification simulation has lower static stability in the mid-troposphere at high latitudes which means for the same forcing there will be more ascent and thus it will be easier for ETCs to develop. Secondly, as the polar regions are warmer, diabatic heating is likely larger in the polar amplification simulation which can increase the intensity of the ETCs beyond the threshold value of $1 \times 10^{-1}$ used to detect an ETC. This indicates that high latitude cyclones may become more common in the future.

The third main result from the ETC clustering was that the relationship between maximum vorticity and precipitation depended strongly on the type of ETC. The strongest relationship occurred for the warm front ETCs and this result is robust across all three experiments. These ETCs were moderately strong, resembled the Norwegian cyclone model, and occurred more poleward than both the cold front or centre ETCs. However, the warm front ETCs produced less precipitation than the cold front ETCs for ETCs of the same maximum vorticity.

A strong correlation was also found between maximum vorticity and precipitation for the cold front ETCs, but this was slightly weaker, and had more spread, than what was found for the warm front ETCs. The cold front ETCs were, on average, the strongest ETCs in terms of vorticity, ascent, and precipitation. The weak ETC cluster had a notably smaller correlation between ETC precipitation and maximum vorticity in all experiments (correlation coefficients range from 0.241 to 0.298). These ETCs occurred at high latitudes and therefore we concluded that the ETC precipitation was limited by the lack of moisture and hence that the dynamic intensity of the ETC had little control on the ETC precipitation.

The weakest of all correlations between the ETC precipitation and vorticity occurred for the cyclone centre ETC (correlation coefficients range from 0.098 to 0.190). Binder et al. (2016) classified cyclones based on their deepening rate and warm conveyor belt intensity and one of their subsets, C2b, has a strong warm conveyor belt and low-level PV anomaly but does not deepen rapidly, which Binder et al. (2016) attribute to the lack of upper level forcing. This C2b composite cyclone also has precipitation located over a relatively small area near the cyclone centre and thus resembles our weak cyclone composite. Therefore, we hypothesise that the poor correlation we found between ETC precipitation and vorticity in the cyclone centre cyclone was due to the lack of strong upper-level forcing for these ETCs, as suggested by Binder et al. (2016).

The final notable result from the clustering was that how the maximum vorticity of the ETCs responsed to uniform warming depends on the type of ETC. The cold front ETCs experienced an increase in their maximum vorticity with uniform warming whereas both the warm front and cyclone centre ETCs showed no change in the maximum vorticity and the maximum vorticity of the weak ETC decreased. Hence this is consistent with the overall picture, when all ETCs are considered together that there were no changes in ETC intensity, however it does suggest that the dynamical intensity of certain types of ETCs may respond

more to climate change than others. This result is also in agreement with previous studies (e.g. Priestley and Catto, 2022) which show that extremes respond differently to warming as the cold front ETC includes the strongest ETCs.

     Our study is unique in some aspects. First, previous studies have tended to relate ETC precipitation to the ETC winds whereas we used the maximum vorticity as identified by TRACK. This approach was selected as maximum vorticity is easily available from climate model simulations and is often the variable that is used when the question of how will the intensity

of ETCs change in the future is addressed. Our study also differs from many previous studies in that we considered ETCs of all strengths and not just the extremes in terms of either their maximum vorticity (as was done by Sinclair et al. (2020)) or precipitation. This does make a difference to how future changes are seen (Priestley and Catto, 2022). However, our study does highlight that despite small changes in the strength of the cyclones, the precipitation increases are large, indicating potential future increases in flooding associated with cyclones.

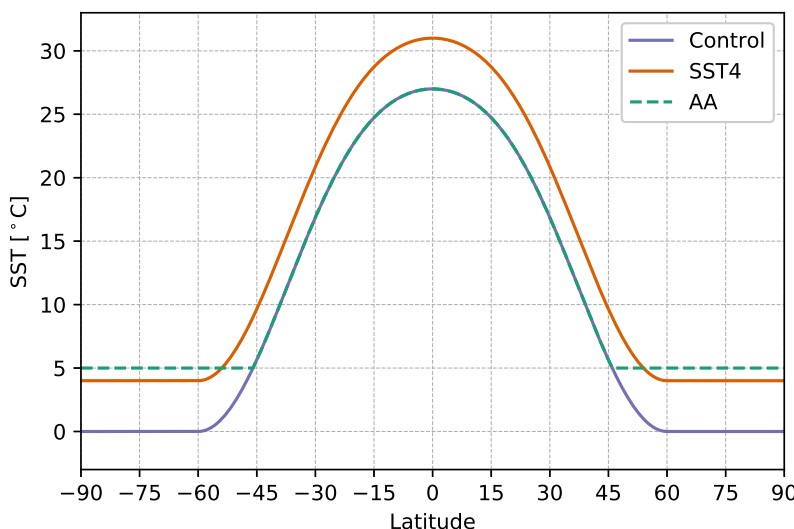

**Figure 1.** Sea surface temperature distribution as a function of latitude for the three experiments. The control simulation is shown in purple which has the QObs SST distribution, SST4 (orange) is uniform warming and AA (dashed green) is warmed poles.

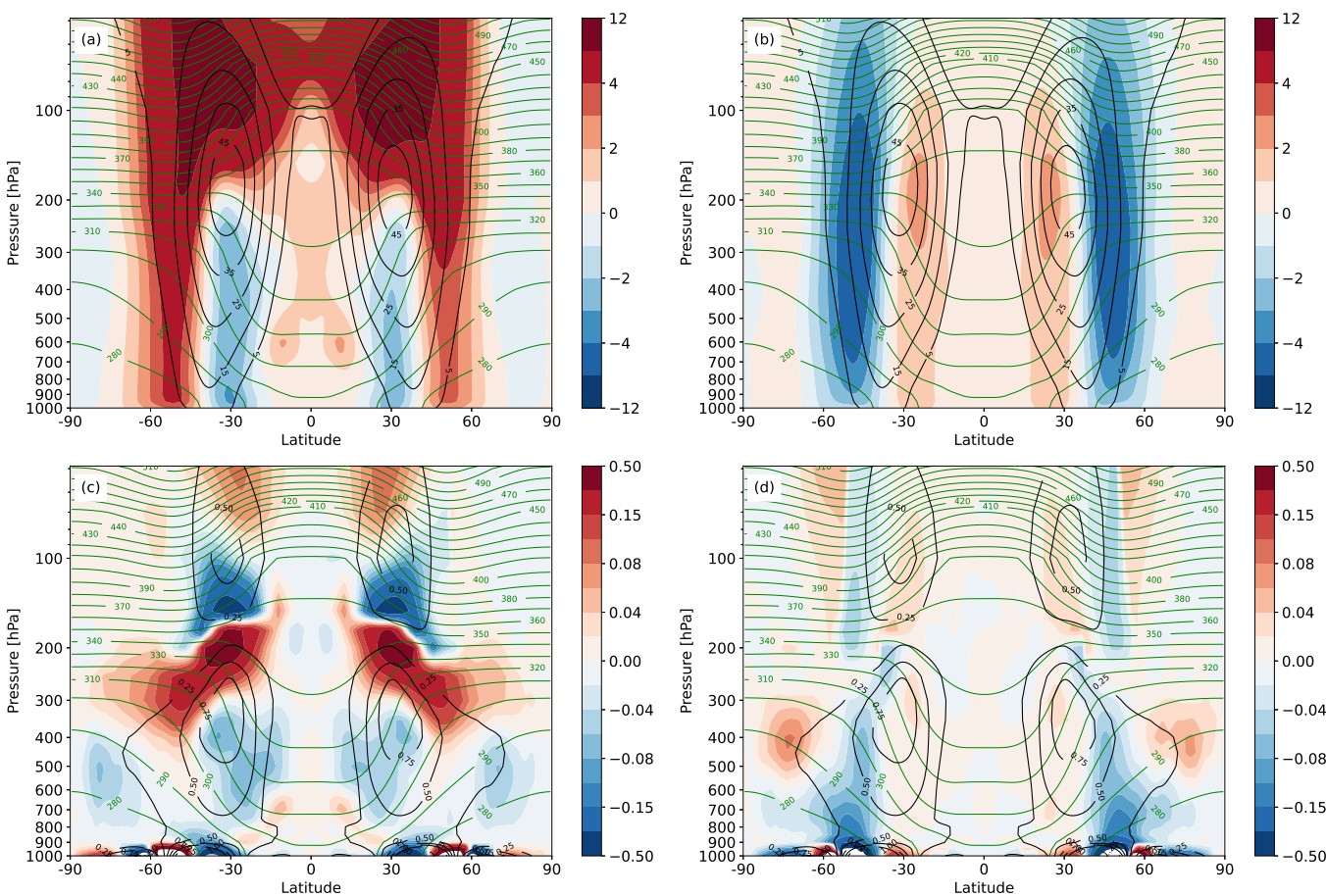

**Figure 2.** Zonal and time mean fields averaged over 10 years of simulation. (a) Potential temperature (green contours, contour interval 10K) and the zonal wind speed (black contours, contour interval 10 m s$^{-1}$) in the control simulation. Shading shows the difference in the zonal wind speed between the control and SST4 simulation (SST4 - control). Panel (b) is the same as (a) except the shading shows the difference between the control and AA. Panel (c) shows the potential temperature (green contours, contour interval 10K) and the Eady growth rate (black contours, contour interval 0.25 d$^{-1}$) in the control simulation. Shading shows the difference in the Eady growth rate between the control and SST4 simulation. Panel (d) is the same as (c) except the difference is between the control and the AA experiment. Note the non-linear colorbars.

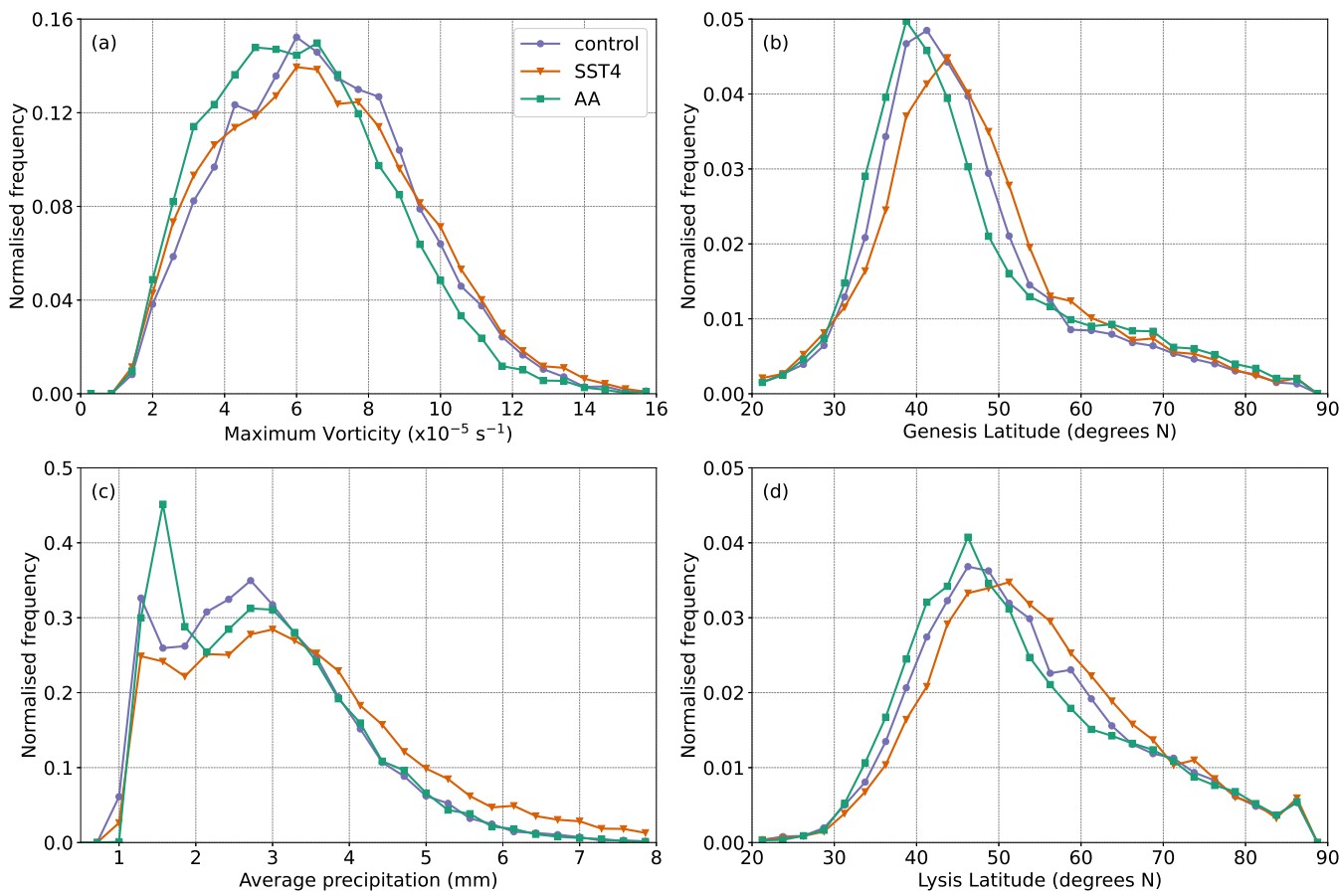

**Figure 3.** Probability density distributions of the (a) maximum relative vorticity (T42 values), (b) genesis latitude, (c) average ETC precipitation 12 hours before the time of maximum intensity and (d) lysis latitude for each experiment. Cyclones are only included if they exist 24 hours before the time of maximum vorticity and reach their maximum vorticity poleward of 30°N.

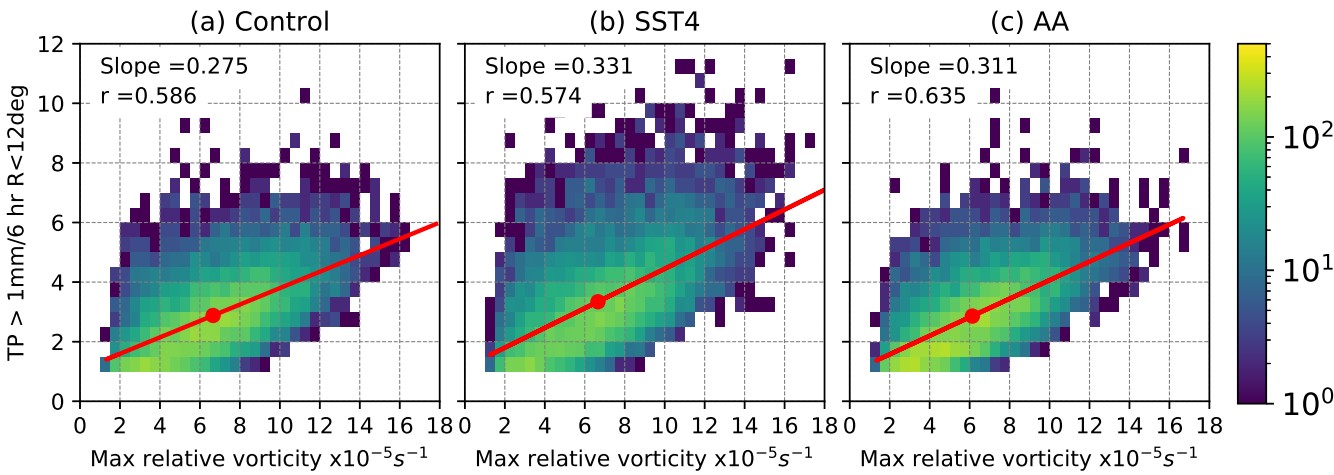

**Figure 4.** Two-dimensional histograms showing the relation between maximum relative vorticity (T42 values) and ETC-related precipitation 12 hours before the time of maximum relative vorticity. Precipitation is the area average, averaged over all points within a 12 degree radius of the ETC centre where the rain rate exceeds 1 mm / 6 hr. Only ETCs which exist at -24 hr and have their maximum vorticity north of $30°$N are included. The slope values have units of mm $(6 \text{ hr})^{-1}$ / $10^{-5}$ $s^{-1}$

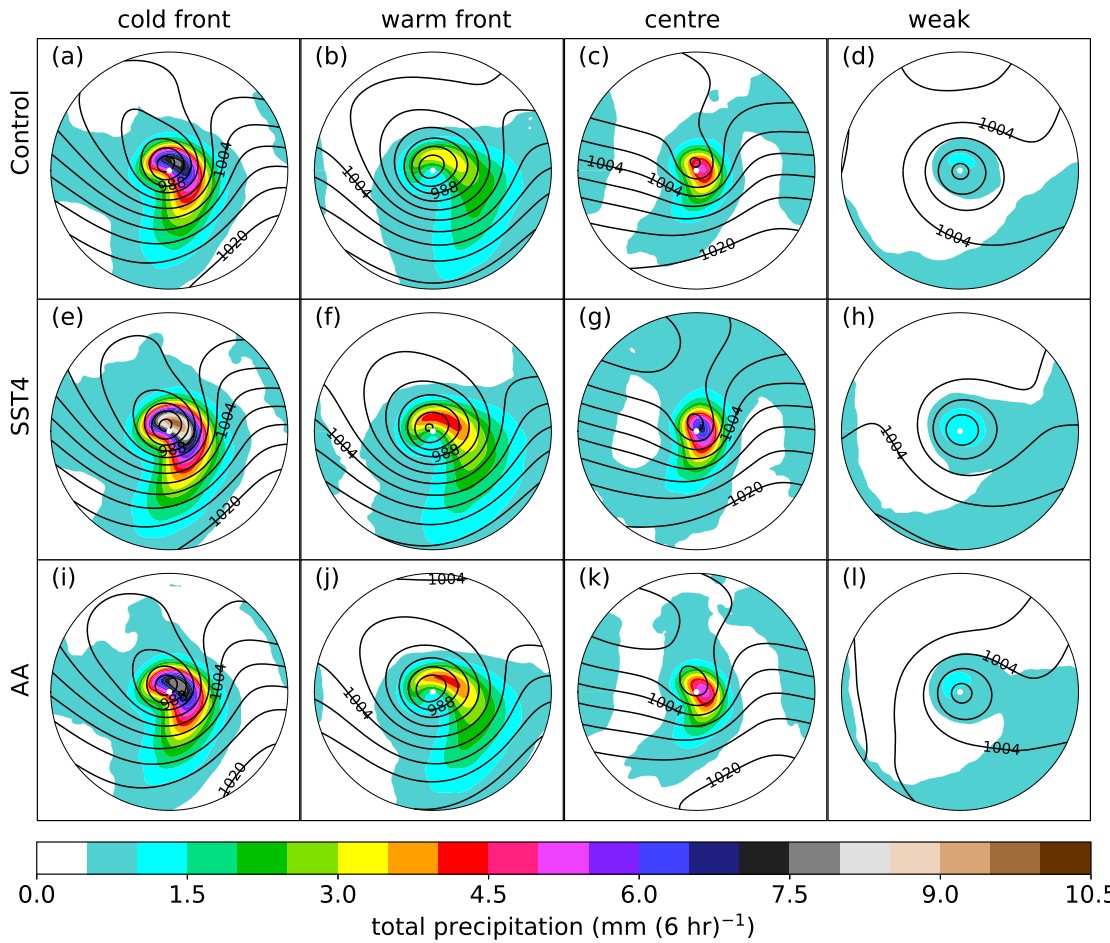

**Figure 5.** Composite mean of the total precipitation (shading, mm / 6 hr) and the mean sea level pressure (grey contours, every 4 hPa) 12 hours before the time of the maximum vorticity. Different columns show different clusters and different rows show different experiments.

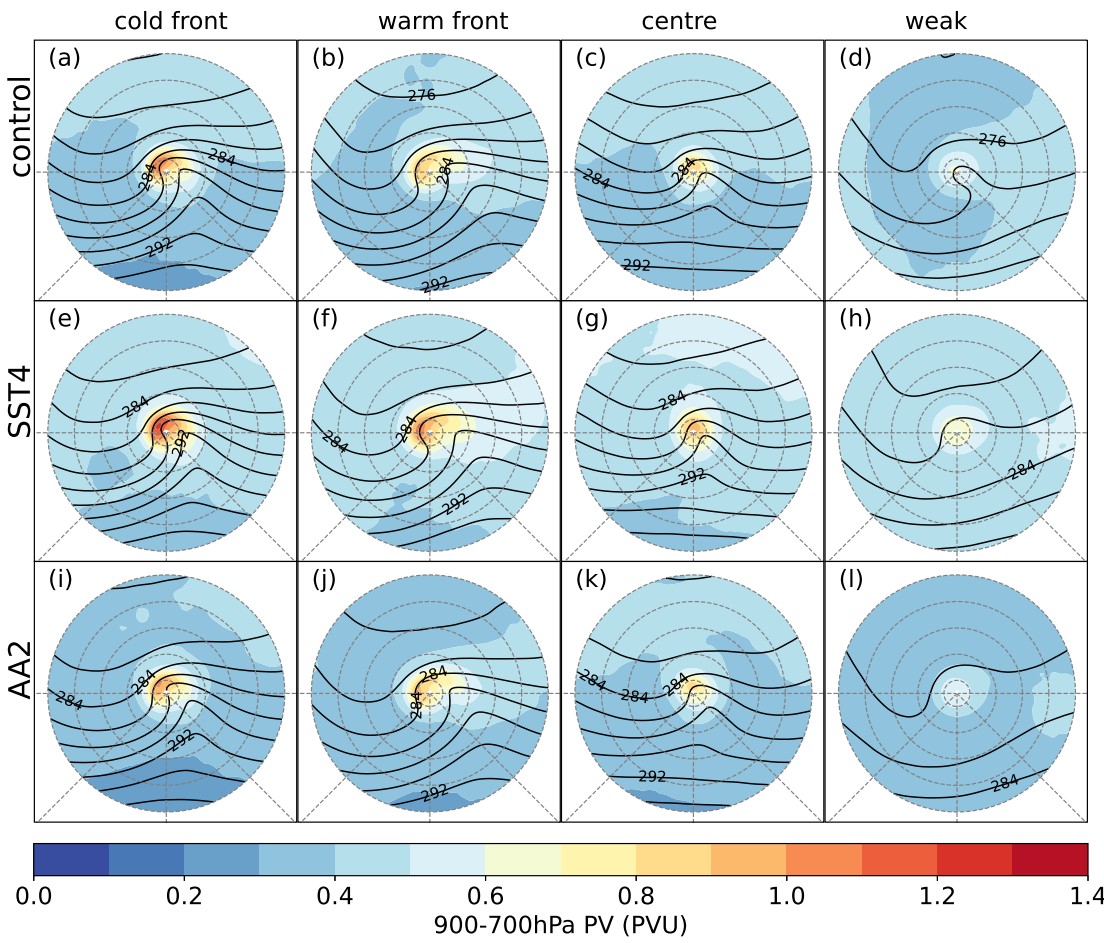

**Figure 6.** Composite mean of the 900-700 hPa layer averaged potential vorticity (shading, PVU) and the 850-hPa potential temperature (black contours, every 2K) 12 hours before the time of the maximum vorticity. Different columns show different clusters and different rows show different experiments.

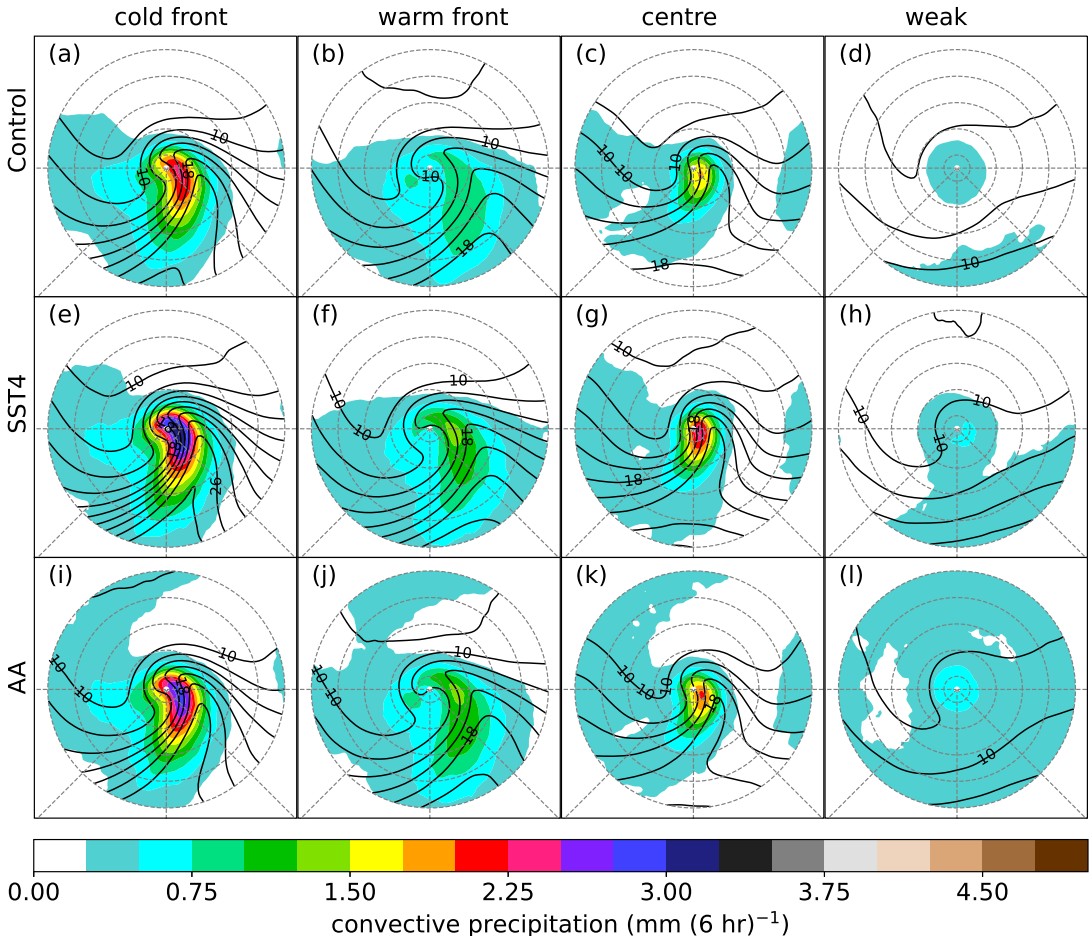

**Figure 7.** Composite mean of the convective precipitation (shading, mm / 6 hr) and the total column water vapour (black contours, every 2 g kg$^{-1}$) 12 hours before the time of the maximum vorticity. Different columns show different clusters and different rows show different experiments.

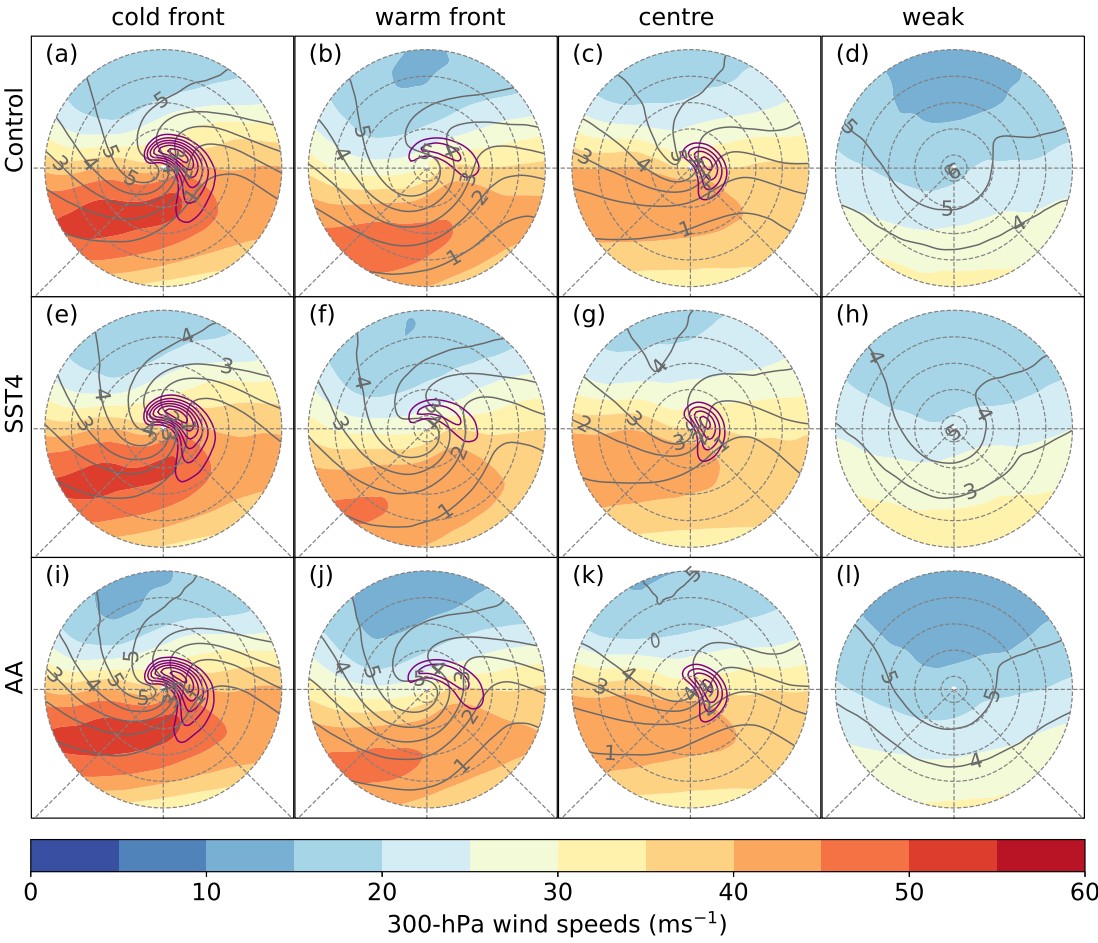

**Figure 8.** Composite mean of the 300-hPa wind speed (shading, m s$^{-1}$), potential vorticity on the 315K surface (grey contours, every 1 PVU) and the 700-hPa vertical velocity (solid purple contours, only ascent shown - (first contour is -0.2 Pa s$^{-1}$, contour interval of 0.1 Pa s$^{-1}$) 12 hours before the time of the maximum vorticity. Different columns show different clusters and different rows show different experiments.

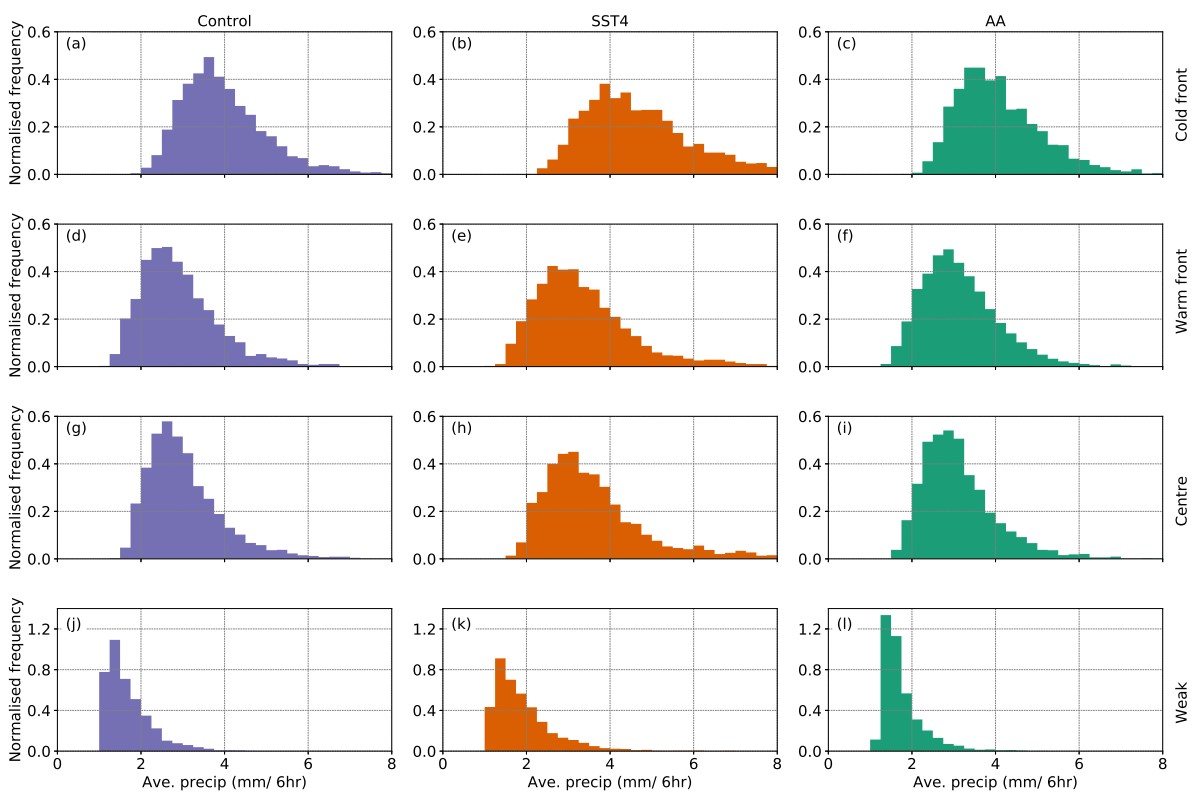

**Figure 9.** Probability density distributions of the average ETC precipitation 12 hours before the time of maximum intensity for each cluster (different rows) and each experiment (different columns). Cyclones are only included if they exist 24 hours before the time of maximum vorticity and reach their maximum vorticity poleward of $30°$N.

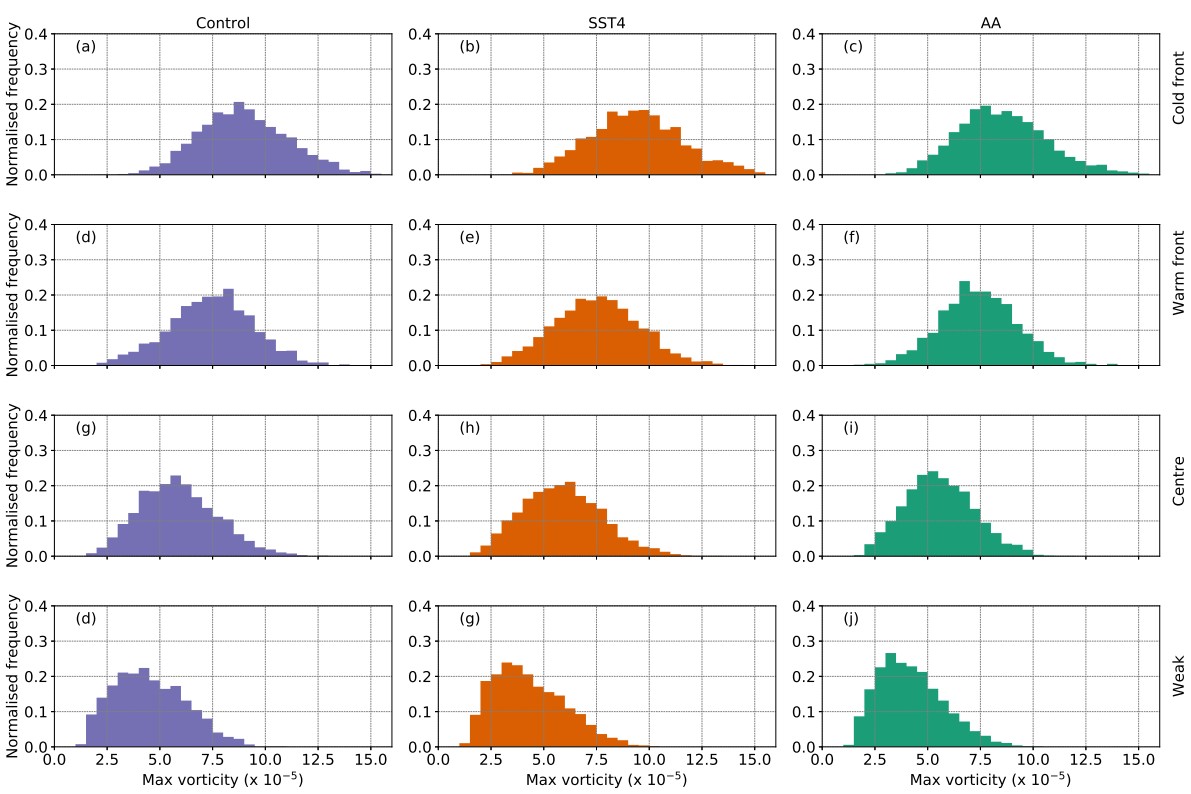

**Figure 10.** Probability density distributions of the maximum 850-hPa vorticity (T42 values) for each cluster (different rows) and each experiment (different columns). Cyclones are only included if they exist 24 hours before the time of maximum vorticity and reach their maximum vorticity poleward of 30°N.

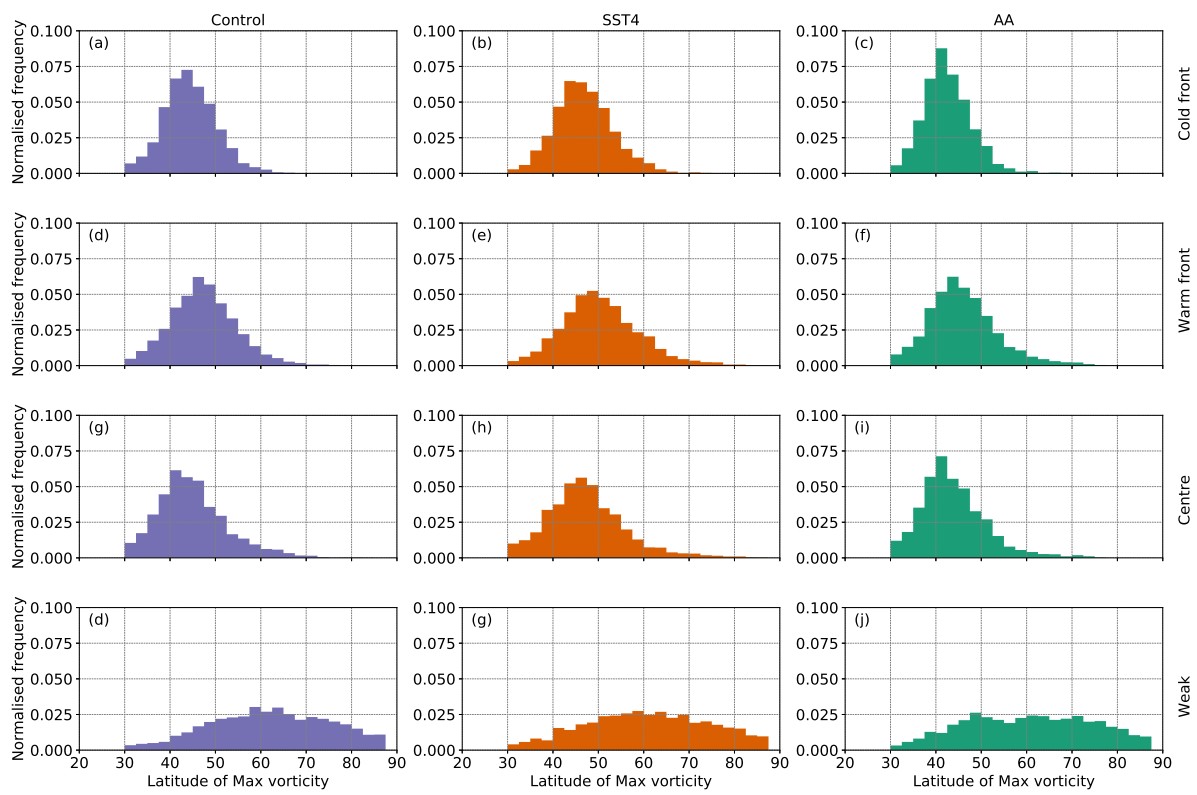

**Figure 11.** Probability density distributions of the latitude of the the maximum 850-hPa vorticity for each cluster (different rows) and each experiment (different columns). Cyclones are only included if they exist 24 hours before the time of maximum vorticity and reach their maximum vorticity poleward of 30°N.

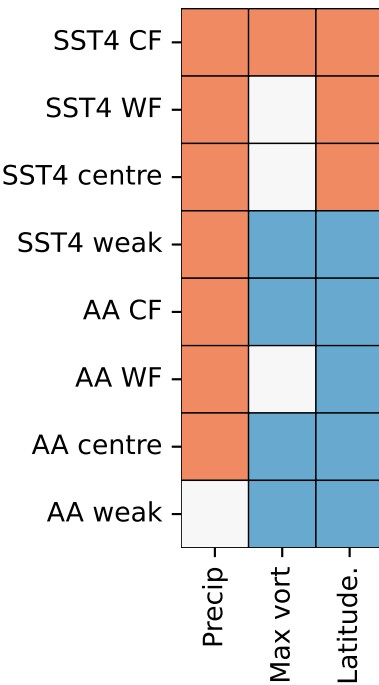

**Figure 12.** Summary of statistically significant differences between each cluster's precipitation, maximum vorticity, and latitudes of maximum vorticity and the corresponding cluster in the control simulation. Orange (blue) indicates that the cluster as labelled on the y-axis has a statistically larger (smaller) value than in the corresponding cluster in the control simulation. White boxes mean there is no statistically different between the cluster labelled on the y-axis and the corresponding cluster in the control simulation. CF indicates the cold front ETC and WF the warm front ETC.

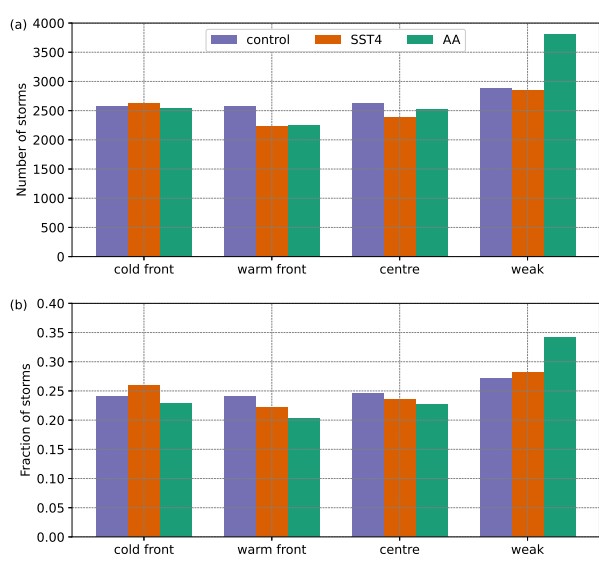

**Figure 13.** (a) The total number of ETCs in each cluster for each experiment and (b) the relative fraction of ETCs in each cluster for each experiment.

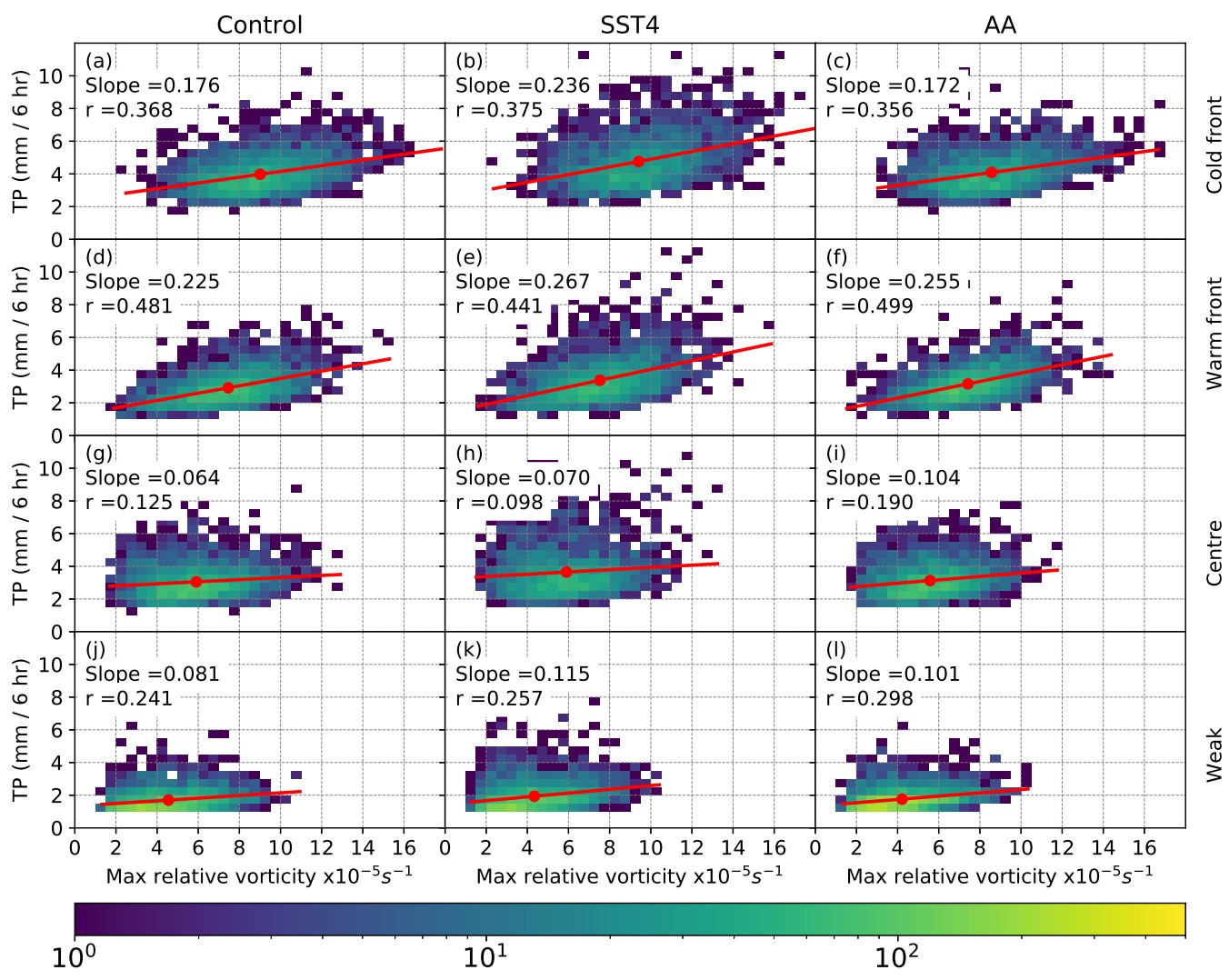

**Figure 14.** Two-dimensional histograms showing the relation between maximum relative vorticity (T42 values) and ETC-related precipitation 12 hours before the time of maximum relative vorticity for each simulation and each cluster. Precipitation is the area average, averaged over all points with a 12 degree radius of the ETC centre where the rain rate exceeded 1 mm / 6 hr. Only ETCs which exist at -24 hr and have their maximum vorticity north of 30°N are included.

**Table 1.** Mean temperature (K) and mean precipitation (mm per day) from all experiments. Values are area / global averages, not ETC-related values.

| Diagnostic | Control | SST4 | AA |
|---|---|---|---|
| Global mean 2-m temperature | 286.7 | 290.8 | 287.8 |
| Global mean precipitation | 3.18 | 3.63 | 3.17 |
| 25N - 70N mean 2-m temperature | 280.1 | 284.2 | 281.7 |
| 25N - 70N mean precipitation | 2.56 | 2.90 | 2.62 |

**Table 2.** ETC statistics from the control, SST4 and AA experiments. Relative vorticity values have units of $\times 10^{-5}\text{s}^{-1}$ and latitudes are degrees North. ETCs are only included if they exist 24 hours before time of maximum vorticity and reach their maximum vorticity north of $30°\text{N}$

| Diagnostic | Control | SST4 | AA |
|---|---|---|---|
| Number of ETCs | 10669 | 10121 | 11139 |
| Median number of ETCs per year $\pm$ 1 std | $1133 \pm 16.4$ | $1134 \pm 24.3$ | $1160 \pm 14.2$ |
| Mean maximum 850-hPa vorticity | 6.66 | 6.66 | 6.14 |
| Median maximum 850-hPa vorticity | 6.53 | 6.50 | 5.97 |
| Standard deviation of maximum 850-hPa vorticity | 2.56 | 2.71 | 2.45 |
| Percentage of cyclones with max vort $> 10\times 10^{-5}\text{s}^{-1}$ | 10.3% | 12.1% | 6.7% |
| median genesis latitude | 43.6 | 45.3 | 42.7 |
| median latitude of maximum vorticity | 47.3 | 49.6 | 46.2 |
| median lysis latitude | 51.2 | 53.1 | 49.8 |

**Table 3.** Slope values (units of $(mm\ (6\ hr)^{-1}\ /\ 10^{-5}\ s^{-1})$) and correlations coefficients between maximum vorticity (T42 values) and ETC precipitation at different offset times for each experiment

| offset time | Control | | SST4 | | AA | |
|---|---|---|---|---|---|---|
| | slope | r | slope | r | slope | r |
| -72 hr | 0.204 | 0.363 | 0.241 | 0.352 | 0.202 | 0.352 |
| -48 hr | 0.247 | 0.474 | 0.290 | 0.459 | 0.263 | 0.526 |
| -24 hr | 0.261 | 0.566 | 0.319 | 0.561, | 0.298 | 0.615 |
| -12 hr | 0.275 | 0.586 | 0.331 | 0.574 | 0.311 | 0.635 |
| 0 hr | 0.227 | 0.504 | 0.258 | 0.483 | 0.254 | 0.561 |
| +24 hr | 0.108 | 0.282 | 0.121 | 0.278 | 0.129 | 0.354 |

*Code availability.* OpenIFS is available under license from the European Centre for Medium Range Weather Forecasting (ECMWF). See https://confluence.ecmwf.int/display/OIFS for more details. TRACK-1.5.2 is available at https://gitlab.act.reading.ac.uk/track/track

*Author contributions.* VAS and JLC designed the study. VAS performed the experiments, and did most of the data analysis and writing. Both authors contributed equally to the interpretation of the results.

*Competing interests.* There are no competing interests

*Acknowledgements.* This research was supported by the Academy of Finland (grant no 338615), Natural Environment Research Council (NERC) grant NE/V004166/1, and Natural Environment Research Council (NERC) grant 625 NE/S004645/1. The authors wish to acknowledge CSC – IT Center for Science, Finland, for computational resources and ECMWF for providing the OpenIFS model. We thank Kevin Hodges for providing the cyclone tracking code TRACK, and Helen Dacre for providing an initial version of the cyclone composite code.

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
