# Peer review of "The relationship between extra-tropical cyclone intensity and precipitation in idealised current and future climates"

_Weather and Climate Dynamics, 2022_

## Referee Comment (RC2)

**Review of "The relationship between extra-tropical cyclone intensity and precipitation in idealised current and future climates" by Sinclair and Catto submitted to Weather and Climate Dynamics**

**General comments**

This study investigates the relationship between extratropical cyclone intensity and precipitation in (aquaplanet simulations of) current and future climates – a topic that is still not fully understood and thus important to be investigated. The main novelty is that the authors carefully look at this relationship in four interesting groups of cyclones, which have been obtained by clustering the precipitation fields of the cyclones, and for all these different idealized climates. The manuscript is very well written over large parts, the figures are mostly very clear and illustrative, and the use of the different methods and tools are well introduced and motivated. While I found many findings and sensitivities of these cyclone types interesting and inspiring, I think the study gives too much the impression that cyclone intensity is only driven by latent heating, because it hardly mentions all the other dynamical drivers, which, however, are crucial to be included when considering cyclones in a future climate. Therefore, I suggest to give more insight into these aspects, which elaborate more in first two major comments. Note that I did not read the first review, which has been posted a week ago, until finishing my own review below.

**Major comments**

In my opinion, the manuscript lacks a discussion on how the three-dimensional background state in the control experiment is characterized and how it changes in the SST4 and AA experiments. Although the focus of this study is on changes in ETC precipitation, some rather strong conclusions are drawn with respect to the influence of precipitation and thus diabatic heating on ETC intensity by only considering the linear relationship between the two variables but without consulting the potential role of all other (dry-)dynamical drivers (apart from the upper-level jet). For instance, O'Gorman and Schneider (2008) showed how eddy kinetic energy (and thus ETC intensity) in aquaplanet climates scales linearly with mean available potential energy (MAPE), which is proportional to the vertically integrated meridional temperature gradient, static stability, and the depth of the extratropical troposphere. I assume that in your climate change experiments these individual measures change in different ways and thereby influence MAPE and thus ETC intensity in various, partly counteracting ways – as it is the case in full GCM projections. I think, knowing about these changes, at least qualitatively, would be important to understand the changes in ETC intensity in your study more holistically. For instance, one of your conclusions is (this appears at several locations of the manuscripts): "In our warmer experiments, precipitation strongly increases but there is not any large increase in the intensity of ETCs i.e. the slope is larger in SST4 than in the control. This implies that the increased diabatic heating does not feedback strongly onto the intensity of the ETCs in these experiments." However, other studies have shown that increased latent heating / precipitation itself actually does or would lead to a further increase in ETC intensity via the well-known mechanisms, but this is balanced by the counteracting effect of reduced MAPE (e.g., Büeler and Pfahl, 2019). So, couldn't it be that your increase in precipitation in SST4 itself actually does increase the vorticity strongly linearly, but the increase is partly dampened by other counteracting factors not seen from this linear relationship, which is why the slope changes?

I am not very familiar with the full-GCM scenarios regarding Arctic amplification. Therefore, I wonder how realistic the prescribed SST setup in your AA experiment is. When I make up the meridional gradient in my mind from your SST curves in Fig. 1, it seems that in your AA experiment the lower-tropospheric meridional temperature gradient does not really decrease but the zone of the strong gradient

is just narrowed and shifted equatorward. Is that correct? And, if yes, don't the full-GCM polar amplification scenarios rather show a flattening of the curve and thus really a reduction of the gradient? I am not saying that your AA setup should be as realistic as possible, but I think you should better discuss how your setup influences the background state (because change of ETC intensity seem to be quite sensitive to how exactly baroclinicity, both in the lower and upper troposphere, and temperature/moisture change; e.g., Tierney et al. 2018) and how it compares to the "real" polar amplification.

Did you also investigate the whole cyclone life cycle characteristics in the three experiments in more detail? For instance, did you look at potential changes in cyclone lifetime, intensification rates, propagation speed, or accumulated precipitation over their lifetime? Thinking of impacts, particularly the latter two measures might be interesting, considering the fact that extreme precipitation is often caused by very stationary cyclones. Of course, this is often related to topography, but it might still be interesting to see whether these characteristics differ even in aquaplanets. This comment is more meant as a nice-to-have and is not crucial for the paper.

**Minor comments**

L11: What do you mean by "differing slopes"?

L79-80: Do you refer to the frequency distribution of ETC intensities here? In other words, you say that the relative frequencies of different intensities do not change, right? You could write that more specifically.

L136: What exactly does "QObs" mean?

L150: Out of curiosity, are there many ETCs that do not at all propagate in your aquaplanet simulations (i.e. the ones you filter out)? I would assume, given the fact that no topography is there to generate standing waves and no land is there to produce, for instance, heat lows, most ETCs propagate, right?

L213: I assume it's global mean surface temperature, right?

L224-225: I think it's interesting that you find most ETCs in the AA experiment, in which, considering its background state, less eddies would be needed to transport heat poleward. Do you think the higher number of ETCs is just (over-)compensated by their weaker intensity (which would result in an equal or even weaker poleward heat transport)?

L236-237: What about the latitude distribution during maximum intensity? Is there also an equatorward shift in AA, as for genesis and lysis? Does this shift result from the equatorward shift of the baroclinic zone in this experiment? And how does this response compare to polar amplification scenarios of full GCMs? (see also my second major comment)

L258: The last part of this long sentence ("ETCs with the same intensity...") is not clear to me – can you rephrase?

L274: What does the "s" mean in "(mm / 6 hr) s"?

L277-279: "Thus, we can conclude that there is a stronger relationship between maximum vorticity and precipitation in the SST4 experiment than in both the control or the AA experiment and also that the AA experiment has a stronger relation that the control." I think this statement is misleading, because a

larger slope does not imply a stronger linear relationship, right? What determines the strength of the linear relationship is the r-value, which is actually smallest in SST4, largest in AA, and in between in Control. Hence, strictly speaking, your sentence above is not correct. However, as you explain in the subsequent sentence, it is about how much precipitation changes with a given vorticity change. So I would just rephrase the sentence above somehow (plus at a few other locations in the manuscript, where you write about the "strength of the relationship" referring to the slope).

L279-282: "This means that for the same increase in maximum vorticity, precipitation increases more in the SST4 and AA experiments compared to the control. Since the SST4 and AA experiments are both warmer than the control experiment this is to be expected based on the Clausius-Clapeyron relationship between temperature and vapour pressure." Is this really that obvious / to be expected? If only the C-C relationship would matter, then also the weak cyclones (low relative vorticity) would need to have equally enhanced precipitation, which would imply that the slope should stay the same, but the intersect would change, right? So, maybe one could also interpret the increased slope such that the stronger cyclones are better able to convert the enhanced moisture content into enhanced precipitation than the weaker cyclones (likely due to the stronger dynamically induced forcing). Does that make sense?

Fig. 4 (and related figures): Could you change the color of the SLP contours? It's very hard to distinguish them from the black TH@850 contours.

L288: Did you consider just showing equivalent potential temperature instead of potential temperature and total column water? It might be the better variable to show temperature distribution, the sharpness of the fronts, and the moisture availability at once. But it's a matter of taste, I guess.

L293: It's interesting to see these four clusters. Do you know about similar clusters in reanalysis? If yes, do they look similar and could you refer to these studies?

Fig. 6: I very much like to see the upper-level structure of the different clusters. Just for my understanding, could it be that some jet maxima we see in this figure are related to the subtropical jet? This also relates a bit to my first major comment regarding how the background state looks like on average.

L305-306: You might also add here that the upper-level PV values downstream are lowest in this cluster, probably because of the strongest diabatic erosion of PV downstream due to the WCB ascent.

L348-349: Do you know why the cold front ETCs (in the control) are more south than the warm front ETCs? Is this also the case when comparing climatologies of Norwegian cyclones with Shapiro-Keyser cyclones? I assume thanks to this fact, the cold front ETCs can draw more from the subtropical moisture reservoir and thus become more intense with more precipitation than the warm front ETCs, right?

L352: I think there is something off with the sentence "The mean latitude that ETCs in the centre ETC cluster reach their maximum vorticity…"

L367-368: Regarding the fact that the tropopause level might change in the different simulations (see also my first major comment), did you also look at other vertical levels when investigating upper-level PV?

L397: Which one is the "small cluster" you mention here?

L415-419: I find the strongest correlation between vorticity and precipitation in the warm front cluster interesting, but I think I don't understand the reason. I would have expected it to be stronger in the cold front cluster. Do you have an explanation for this?

L507-510: "The cold front ETC sees an increase in its maximum vorticity with uniform warming whereas both the warm front and cyclone centre ETCs show no change in the maximum vorticity and the maximum vorticity of the weak ETC decreases." Isn't this somehow contradictory to the aforementioned fact that the vorticity-precipitation relationship is strongest and with the largest slope in the warm front ETCs, which get much more precipitation in the warm experiments? Or does this just show that we cannot understand everything from just looking at these two variables (see also my first major comment)?

**Typos / suggestions for rephrasing**

L32: Replace "Even" with "Already"?

L33: Comma after "ascending"

L40: Replace the second "using" with "based on"

L70: Rephrase to "that the number of ETCs associated with extreme precipitation may triple"

L72: Comma or semicolon after "simulation"

L95: Delete "will be on"

L108: "setup" rather than "set up"?

L126: Write consistently either "aquaplanet" or "aqua-planet".

L141: "those previous simulations"

L167: "due east" -> "to the east"?

L238: Comma after "Lastly"

L265: Rephrase to "The same as Figure 3 but for precipitation 24 and 0 hours before the time of maximum vorticity are shown in the Supplementary material."

L279: "… stronger relation than the control."

L286: "… than in other simulations."

L314-315: Delete "associated with them"

L327: Delete "associated with it"

L342: "whereas the weak ETCs have very different shaped distributions"

L354: Delete "associated with them"

L389: Delete "associated with then"

L396: ")" after "Fig. 10"

L397: "with the weak ETC cluster"

L425: Delete "associated with them"

L493: Typo in "vorticity"

**References**

Büeler, D., and Pfahl, S. (2019). Potential Vorticity Diagnostics to Quantify Effects of Latent Heating in Extratropical Cyclones. Part II: Application to Idealized Climate Change Simulations. Journal of the Atmospheric Sciences 76, 1885-1902, https://doi.org/10.1175/JAS-D-18-0342.1

O'Gorman, P. A., and Schneider, T. (2008). Energy of Midlatitude Transient Eddies in Idealized Simulations of Changed Climates. Journal of Climate 21, 5797-5806, https://doi.org/10.1175/2008JCLI2099.1

Tierney, G., Posselt, D.J. and Booth, J.F. (2018). An examination of extratropical cyclone response to changes in baroclinicity and temperature in an idealized environment. Climate Dynamics 51, 3829–3846,. https://doi.org/10.1007/s00382-018-4115-5

---

## Author Comment (AC1)

**Response to reviewers - The relationship between extra-tropical cyclone intensity and precipitation in idealised current and future climates**

Victoria Sinclair and Jennifer Catto

February 24, 2023

**1 Reviewer 1**

**1.1 Summary**

In this study, the authors perform global aquaplanet simulations of extratropical cyclones (ETCs) in current and warmed climates. They use a feature-tracking algorithm to diagnose the ETC life cycles and a clustering algorithm to group the ETCs according to their precipitation structures. Through this approach, they evaluate differences between four clusters in both current and future climate, particularly in terms of the relationship between relative vorticity and precipitation amount. They conclude that the types of ETCs for which this relationship is the strongest are the ones where most of the precipitation is concentrated along the warm front. They also argue that stronger diabatic heating in warmer climates, owing to increased atmospheric moisture content, does not feed back onto the dynamical intensity of the storm.

Altogether, I found this manuscript to contain interesting analyses that make excellent use of various meteorological tools (GCMs, cyclone tracking, and clustering) to quantify ETC variability in current and future climates. For the most part, the results are presented clearly and supported by figures and tables. Given that this study is mostly diagnostic and descriptive, with little dynamical interpretation of the results, such interpretation would be needed to translate this result into improved dynamical understanding. But I can accept that a proper interpretation can wait for future work.

For the present work, I believe that there is scope for some improvement, to address lack of clarity or conclusions that, at least in my view, are not justified based on the data presented. In the following, I highlight those areas in an effort to stimulate improvements to the manuscript.

*We thank the reviewer for the overall positive assessment of our manuscript and for the constructive comments. We have copied all of the reviewers comments below in black and have added our response to all points in blue.*

**Major comment**

1. As mentioned above, the authors conclude that "increased precipitation in the warmer simulations does not feed back, via diabatic heating and potential vorticity anomalies, onto the

dynamical intensity of the ETCs." This is a rather provocative conclusion that isn't based on a very rigorous analysis. They argue on P. 9 that, if diabatic heating did feed back on the dynamics, that "similar slopes would be found in all experiments."

Although I agree that the slope would be shallower (and more like the control) if there was a positive feedback of diabatic heating on cyclone intensity, how can you conclude that the slope would be the same? This would seem to imply that diabatic heating is the only factor affecting cyclone strength. Is it possible that other factors could also influence this relationship, and that these factors may also change from the control to the SST4 cases? Because the only input that changes is the SST, one may conclude that the only dynamical difference is the diabatic heating. That may be the case, but without a deeper analysis I'm not entirely sure of that. Therefore, I think the authors should reconsider the strength of this conclusion, or find a stronger justification for it.

We did not mean to imply that diabatic heating is the only factor influencing the strength of the cyclones, however, we hypothesised that it would be an important aspect given the degree of warming imposed on both of the sensitivity experiments. Our hypothesis also was that if there was a strong diabatic feedback, then we would expect both precipitation and vorticity to increase and hence that there would be no dramatic changes to the slope of the regression line between these two variables. Vice-versa, if there was no feedback or a weak feedback, precipitation would increase more dramatically than vorticity and hence the slope would increase. We acknowledge that this situation - an increase in precipitation but no increase in vorticity could also occur due to decreases in the large-scale baroclinicity weakening the vorticity of the cyclones and hence acting to mask / counter-act the impact of diabatic feedback. We have revised the text on page 9 and elsewhere to reflect this.

In addition, as we know that the large-scale baroclinicity and hence the meridional temperature gradient and stability can influence the intensity of extra-tropical cyclones, we have added an additional 4-panel figure to section 4 of the revised manuscript which quantifies how the zonal and time mean zonal wind speed and Eady growth rate differs between the three simulations. We also now include new figures in the supplementary material of how the Brunt Väisälä frequncy and the meridional potential temperature gradient change between the simulations.

Lastly, we have added a new figure showing the 900 - 700 hPa layer averaged potential vorticity in each ETC composite to help interpret the results with regards to how diabatic heating may feed back onto the circulation in the different experiments and in different clusters.

2. Another possible overstatement is in the abstract on L. 14, where the authors conclude that "precipitation patterns associated with ETCs are unlikely to change in the future". The results do suggest that the mean of the different composites will not change in the future, but you haven't looked at higher-order aspects of the distribution (variance, skewness, etc.). Again, while this statement may ultimately be true, I think the level of certainty of the conclusion is a bit greater than the justification provided by the data.

To look at higher order aspects of the distribution in a spatial sense (i.e. on a map similar to what

[Figure]

Figure 1: Normalised distributions of the Euclidean distance between each cyclone allocated to a cluster and the centroid of that cluster for the (a) cold front, (b) warm front, (c) centre and (d) weak clusters. The different experiments are shown in different colours.

is shown in the original Figures 4, 5 and 6) is not possible as, due to the variation in the exact frontal positions and size of the cyclones, this results in a very noisy field where the variance is very large in certain parts of the cyclone. However, to address this concern we computed the Euclidean distance between each member (i.e each cyclone) allocated to a cluster and the centre of that cluster. The distributions of these distances can then be plotted and compared between the same cluster (e.g. the cold front cluster) in different experiments (Figure 1 in this response). Doing so revealed that the distributions of the Euclidean distances are very similar in all experiments for the cold front, warm front, and centre cluster indicating that the variation within these clusters does not change with warming. The results do indicate that for the weak cluster in the Arctic amplification experiment there is more variation than in the control or SST experiments. Figure 1 is now included in the supplementary material along with other figures showing how far each ETC is from the centroid of the clusters which it is not allocated to. These new results are now discussed in the revised manuscript in section 3.3. We also slightly revised the text on line 14 of the abstract to state that the average precipitation patterns are unlikely to change, however, note that as we do not state this with absolute confidence; we write "are unlikely to change".

**Minor comments**

1. P. 1, L. 11: it is unclear what the term "slopes" is referring to. This referred to the slope of the linear regression between maximum vorticity and precipitation. However, we agree that it was

not clear in the abstract so have now revised this.

2. P. 2, L. 45-46: "can potentially feedback onto the intensity of the ETC by enhancing the low-level circulation and thus the intensity of the ETCs.". This is clearly redundant, so delete "and thus the intensity of the ETCs". Also, feedback should be "feed back"
Revised as suggested

3. L. 139: It would have been a bit cleaner to increase the SSTs at higher latitudes by 4C, for consistency with SST4. I'm not saying this needs to be changed, but it makes comparison between SST4 and AA a bit more complicated than it needs to be. It was difficult to design experiments that were both easy to compare, physically sensible and had SST distributions which were a simple analytical function of latitude. Due to computational and time limitations we decided not to change the set-up of the experiments presented here but we have added extra text to section 2.2 of the revised manuscript to better motivate and justify the experiment set up.

4. L. 143-144: What is meant by the vague statement "did not modify the surface pressure"? A bit more specificity is needed. I'm assuming you are referring to accounting for topography, but it would help to clarify that.
This refers to the previous study, Sinclair *et al.* (2020), and yes it is to account for lack of topography in the aqua-planet compared to in the randomly selected real analysis / set of initial conditions which are used to initialise the model. Some additional details have been added here to make it clearer.

5. L. 160-161: "Previous studies have often selected...". You say that previous studies have done something, but then you don't cite any actual studies. Your point would be more convincing if you cited a couple papers to support your claim. We have added three references here.

6. L. 172-174: Another way to evaluate this is to look at the variability of parameters within each composite (i.e., higher-order moments of the distributions). If higher order moments (e.g. variance) are plotted on maps similar to those in the original figures 4, 5 and 6, then there is a very high amount of variance near the fronts and overall these maps look very messy. The distributions of total precipitation shown in the original figure 7 allow a reader to infer roughly the variability of this parameter between the different clusters and different experiments. In addition, we have now analysed the distributions of the Euclidean distance between each cluster member and its cluster centre and now include these results in the paper. (Also see our response to major comment number 2 above.

7. L. 184: I think you should clarify that your interest in grouping ETCs based on precipitation structure (rather than intensity) is limited to the clustering analysis. You are actually very interested in the absolute amounts in later sections, so this can be misconstrued. Thank you. This is a good point and we have now revised this sentence and added additional information to the end of this paragraph.

8. L. 214-215: This comparison is too vague. You say that the global mean precipitation rate in your simulations is larger than the one on the "real Earth". But how much larger? An order of

[Figure]

Figure 2: The number of ETC occurring in each year in each experiment (left) and the number of ETC occurring in each month of each experiment (right). Note that in the right-hand panel it appears that the sample is less than 120 but this is because there are months with exactly the same number of ETCs and hence the points lie directly on-top of each other.

magnitude, 10%, or something in-between? This is useful information for assessing the level of realism of these simulations. Thank you - this is a good addition. According to Hartmann, the global annual precipitation per year is 973 mm which equates to 2.66 mm /day. Therefore, the control aqua planet simulation has 0.52 mm / day more precipitation than the real planet - an increase of 20%. We have revised this part of the manuscript accordingly.

9. L. 224: 10 samples is a very small number, and it likely downplays the statistical significance of these results. Since there's no annual cycle, why is it necessary to make the sampling interval one year? You could choose seasonal means and get many more samples. It would be interesting to see if the significance changes if you change the sampling rate. We repeated this analysis but using 30 day periods, so a sample size of 120. The results are shown in Figure 2 of this response. When the t-tests were performed, we find that increasing the sample size does not change the main conclusion. When comparing the control and SST4 experiments the p-values are 0.848 when n=10 and 0.853 when n=120. When comparing the control and the AA experiments the p-values are 0.00087 when n=10 and 0.0037 when n=120. As this does not alter the results, we did not revise the manuscript regarding this point. Also, doesn't the t-test require specification

of a confidence level (usually 95%), beyond which the results are considered significant? If so, what is that level? Yes, this is the case and we did use a 95% confidence level. This is now added to the text.

10. L. 242-243: I don't follow this text: "The control simulation has more ETCs with weak to moderate precipitation ( 2 - 3 mm / 6 hr) than either the SST4 or AA simulations, potentially as, on average, the control is the coldest simulation.". The "potentially" part is confusing. We wrote "potentially" as this is an hypothesis, we did not prove that this was the case. We have revised this to make it clearer.

11. L. 252-253: The authors discuss a certain analysis here (called ANCOVA) and report the results. But no results are shown. Is this intentional? This is one place where the reader is unable to verify the result. Is there more information available that can make this a bit more convincing? ANCOVA (analysis of covariance) is a statistical test and the output is p-values. We have now added the returned p-values to the revised manuscript.

12. L. 257-258: To help the reader, can you clarify which one is more efficient at producing precipitation for a given relative vorticity? You say there is a difference, but you don't indicate which one is larger. We have revised this sentence to state that ETCs in the AA simulation have statistically significantly more precipitation associated with them: i.e. a ETC in the AA experiment will have more precipitation than an ETC in the control simulation with the same maximum vorticity.

13. L. 287, "This is likely due to the warmer atmosphere (Table 1) and more diabatic heating in this case than the control or AA simulations." I don't understand the authors' logic here. Why does more diabatic heating weaken the relationship between vorticity and precipitation in SST4? Earlier it was noted that diabatic heating does not seem to feed back much on vorticity, but that is an empirical finding that doesn't physically explain the result. Now the authors seem to be attempting to pose a physical explanation, but I'm not sure what that is. In the original line 287 we were attempting to explain why the spread was larger in SST4 compared to the control i.e. why the Pearson's correlation coefficient was smaller. We hypothesised that in a warmer climate precipitation may be more variable potentially due to an increase in convective precipitation. However, based on many of the reviewers comments, we have re-thought this section carefully with the help of the the new figure showing the low-level potential vorticity in each composite. Many parts of section 5, including this line, have now been revised.

14. L. 327-328 and Fig. 4: I can't verify the authors' conclusion here, because the MSLP contours are very difficult to see in Fig. 4. The figure is too busy with numerous contours on it. Are the range rings really necessary? If not, I strongly suggest removing them. We have removed the range rings on this figure and have also now moved the 850-hPa potential temperature contours to the new figure showing the low-level potential vorticity. As a result, we think this figure is now much clearer to read.

15. L. 353-354, "The distributions of latitude of the maximum vorticity also differ in a statistically significant way." It is unclear if you are referring to all ETCs, or just the weak category that you were referring to in the previous sentence. Sorry this was not clear. We have now revised

this sentence to be clear that this referred to the difference between the warm front ETCs and the centre ETCs.

16. L. 368, "The minimum MSLP is 1-3 hPa deeper". Technically, the word "deep" is used to describe the cyclone centre. But here you are referring to the "minimum MSLP", for which the correct comparison word is "smaller" or "larger". We have revised this to read "1-3 hPa lower" which we think is also a correct option.

17. L. 371: I don't think the term "dry dynamics" is relevant when discussing moist midlatitude cyclones. The dynamics are a mixture of dry and moist, and I'm not sure if you can be certain about one or the other at this. We have revised this to say that the increase in precipitation is not directly related to changes in the ascent.

18. L. 372: Here a "weaker jet streak" is mentioned in a paragraph where no figures are cited. At times, the authors do not cite relevant figures and force the reader to do some detective work in attempting to verify their claims. We have added a reference to the original figure 9b and 9f here. We have also checked elsewhere in the manuscript to make sure figures are referred to where necessary.

19. L. 397: This is the first I've heard of a "small" cluster. Which of the four clusters are you referring to here? Sorry this should be the centre ETC - we changed the name of this cluster at some point and this instance was not updated. It is now fixed.

20. Figure 11: Throughout this paper, the cluster discussions have always referred to the four clusters in a certain order. Here, the order has been changed, as warm-front now goes before, rather than after, the cold-front cluster. Unless there is a need to do this, it would make sense to keep the ordering consistent throughout. We have re-made the original Figure 11 so that the cold front cluster is presented first.

21. L. 412: It may also be worth noting here that the increase in weak ETCs does not come at the expense of stronger ones. There are just more cyclones, but most of the extra ones are weak. Thank you for the suggestion. We have added a sentence about this to the revised manuscript.

22. L. 614: This reference to Owen et al. does not provide sufficient information. To what journal is this "in prep" paper being submitted. If you haven't even decided upon that, it's probably premature to be citing it. A conference presentation may work better here. We have removed this reference.

**Technical corrections**

1. L. 95: Delete "will be on" deleted

2. L. 108: set up − > setup corrected

3. L. 109: analysis − > analyse corrected

4. L. 343-345: This sentence would fit better in the subsequent paragraph. In terms of the logical flow, it is currently out of place. We have moved this sentence as suggested.

5. L. 345-349 and Figs. 7 and 8: the panel labelling seems off in these figures and in the citations to them in the text, making it difficult for me to follow the authors' argument. We have revised the references to the figure panels.

6. L. 389: then $->$ them corrected

**Reviewer 2**

**General comments**

This study investigates the relationship between extratropical cyclone intensity and precipitation in (aquaplanet simulations of) current and future climates – a topic that is still not fully understood and thus important to be investigated. The main novelty is that the authors carefully look at this relationship in four interesting groups of cyclones, which have been obtained by clustering the precipitation fields of the cyclones, and for all these different idealized climates. The manuscript is very well written over large parts, the figures are mostly very clear and illustrative, and the use of the different methods and tools are well introduced and motivated. While I found many findings and sensitivities of these cyclone types interesting and inspiring, I think the study gives too much the impression that cyclone intensity is only driven by latent heating, because it hardly mentions all the other dynamical drivers, which, how- ever, are crucial to be included when considering cyclones in a future climate. Therefore, I suggest to give more insight into these aspects, which elaborate more in first two major comments. Note that I did not read the first review, which has been posted a week ago, until finishing my own review below.

> Thank you for the overall positive comments and constructive comments. We address of your details points (copied in black) below. Our responses are shown in blue.

**Major comments**

1. In my opinion, the manuscript lacks a discussion on how the three-dimensional background state in the control experiment is characterized and how it changes in the SST4 and AA experiments. Although the focus of this study is on changes in ETC precipitation, some rather strong conclusions are drawn with respect to the influence of precipitation and thus diabatic heating on ETC intensity by only considering the linear relationship between the two variables but without consulting the potential role of all other (dry-)dynamical drivers (apart from the upper-level jet). For instance, O'Gorman and Schneider (2008) showed how eddy kinetic energy (and thus ETC intensity) in aquaplanet climates scales linearly with mean available potential energy (MAPE), which is proportional to the vertically integrated meridional temperature gradient, static stability, and the depth of the extratropical troposphere. I assume that in your climate change experiments these individual measures change in different ways and thereby influence MAPE and thus ETC intensity in various, partly counteracting ways – as it is the case in full GCM projections. I think, knowing about these changes, at least qualitatively, would be important to understand the changes in ETC intensity in your study more holistically. For instance, one of your conclusions is (this appears at several locations of the manuscripts): "In our warmer experiments, precipitation strongly increases but there is not any large increase in the intensity of ETCs i.e. the slope is larger in SST4 than in the control. This implies that the increased diabatic heating does not feedback strongly onto the intensity of the ETCs in these experiments." However, other studies have shown that increased latent heating / precipitation itself actually does or would lead to a further increase in ETC intensity via the well-known mechanisms, but this is balanced by the counteracting effect of reduced MAPE (e.g., Büeler and Pfahl, 2019). So, couldn't it be that your increase in precipitation in SST4 itself actually does increase the

vorticity strongly linearly, but the increase is partly dampened by other counteracting factors not seen from this linear relationship, which is why the slope changes?

Thank you for bring up this good point. We fully agree that we should also present the large-scale structure of the atmosphere in these experiments. In the revised manuscript, we now include a new 4-panel figure showing vertical cross sections of the zonal and time mean potential temperature, zonal wind speed, and Eady Growth Rate (EGR) in the control simulation and how the zonal wind speed and the EGR change in both the uniform warming and the Arctic amplification experiments. In addition, in the supplementary material, we also show figures of how the meridional potential temperature gradient and the Brunt Väisälä frequency change in both sensitivity experiments. These new figures show that the changes in the location of the jets are in agreement with changes in the ETC genesis, lysis and latitude of maximum intensity which were already presented in the original manuscript: uniform warming leads to a poleward shift in the jet whereas polar warming leads to an equatorward shift. Changes to the EGR are more complex but in both experiments there is a general decrease in the EGR in the low-to-mid troposphere. In the uniform warming experiment this is caused by an increase in stability whereas in the polar amplification experiment the decrease in the low-level meridional temperature gradient is mainly responsible. These results are discussed in more detail in section 4 of the revised version manuscript.

In addition, we also have revised many parts of the manuscript to make it clear that both changes to the large-scale baroclinicity and changes to diabatic heating can influence the maximum vorticity of the cyclones. Specifically, we have revised some of our previously over-stated conclusions that there is no diabatic feedback as it is likely that the decrease in the large-scale baroclinicity acts to mask this impact.

Lastly, to aid the discussion of diabatic heating and how it potentially feeds back onto the strength of the low-level vorticity, we now include a new figure showing the 900 - 700 hPa layer averaged potential vorticity for each ETC composite in each experiments.

2. I am not very familiar with the full-GCM scenarios regarding Arctic amplification. Therefore, I wonder how realistic the prescribed SST setup in your AA experiment is. When I make up the meridional gradient in my mind from your SST curves in Fig. 1, it seems that in your AA experiment the lower-tropospheric meridional temperature gradient does not really decrease but the zone of the strong gradient is just narrowed and shifted equatorward. Is that correct? And, if yes, don't the full-GCM polar amplification scenarios rather show a flattening of the curve and thus really a reduction of the gradient? I am not saying that your AA setup should be as realistic as possible, but I think you should better discuss how your setup influences the background state (because change of ETC intensity seem to be quite sensitive to how exactly baroclinicity, both in the lower and upper troposphere, and temperature/moisture change; e.g., Tierney et al. 2018) and how it compares to the "real" polar amplification.

We now present cross sections of the meridional potential temperature gradient in the control simulation and how this responds in both simulations in a new figure in the supplementary

[Figure]

Figure 3: Absolute values of the meridional potential temperature gradient at 850 hPa in all three experiments.

material. This shows that the maximum magnitude of the temperature gradient does decrease slightly but also that the decrease occurs on the poleward side of the baroclinic zone, meaning that the baroclinic zone becomes narrower. This is more evident in Figure 3 in this response which shows the 850-hPa meridional potential temperature gradient in all simulations.

To address the concern about how realistic our AA simulation is, we would like to re-iterate that we did not aim to simulate a real situation here. We wanted to isolate the impact of warmer poles without modifying the tropics and sub-tropics. This is unrealistic by default as in reality the low and mid latitudes are also warming. In addition, in our aqua-planet set up there is no sea ice - only water - which is also an unrealistic aspect. However, we agree that we could better justify our experiment design and hence have added additional text into section 2.2 in the revised manuscript

3. Did you also investigate the whole cyclone life cycle characteristics in the three experiments in more detail? For instance, did you look at potential changes in cyclone lifetime, intensification rates, propagation speed, or accumulated precipitation over their lifetime? Thinking of impacts, particularly the latter two measures might be interesting, considering the fact that extreme precipitation is often caused by very stationary cyclones. Of course, this is often related to

[Figure]

Figure 4: Duration of extra-tropical cyclones (hrs) in all three experiments

topography, but it might still be interesting to see whether these characteristics differ even in aquaplanets. This comment is more meant as a nice-to-have and is not crucial for the paper. We did compute the cyclone life time which is shown in Figure 4 of this response. There are some small differences, for example, there are slightly more longer lived cyclones in the polar warming (AA) simulation that in the other two simulations. However, overall no major differences were detected so we do not include this in the revised manuscript. Unfortunately we did not compute the other suggested diagnostics and therefore we do not include these in the revised manuscript. This is mainly because they are not directly related to the aim of this current manuscript. However, as the reviewer points out, they are relevant if we want to investigate potential impacts of the storms and therefore these suggested diagnostics would be interesting to compute in the future.

**Minor comments**

,

1. L11: What do you mean by "differing slopes"? This referred to the slope of the linear regression between maximum vorticity and precipitation. However, we agree that it was not clear in the abstract so have now revised this.

2. L79-80: Do you refer to the frequency distribution of ETC intensities here? In other words, you say that the relative frequencies of different intensities do not change, right? You could write that more specifically. Yes, we refer to the frequency distribution of ETC vorticity here - basically the results shown in Figure 4 of Priestley and Catto (2022). We have revised this to make it more explicit.

3. L136: What exactly does "QObs" mean? We use the same terminology / notation as Neale and Hoskins (2000) who write that "The Qobs SST distribution is a simple geometric function closest to the observed zonal mean SST distributions." We have added a few more details to the manuscript, however, please note that many other studies (e.g. Möbis and Stevens (2012), Chen *et al.* (2010)) also use this QObs term.

4. L150: Out of curiosity, are there many ETCs that do not at all propagate in your aquaplanet simulations (i.e. the ones you filter out)? I would assume, given the fact that no topography is there to generate standing waves and no land is there to produce, for instance, heat lows, most ETCs propagate, right? We did all of the filtering at once so it is difficult to determine how many cyclone each threshold / requirement removed. However, we agree with the reviewer's hypothesis that in an aquaplanet it is unlikely that there are a lot of stationary cyclones.

5. L213: I assume it's global mean surface temperature, right? The model output from our OpenIFS simulations is 2-m temperature. In Hartmann (2015) it is stated that the global mean surface temperature is 288K. We have revised the manuscript to be more precise.

6. L224-225: I think it's interesting that you find most ETCs in the AA experiment, in which, considering its background state, less eddies would be needed to transport heat poleward. Do you think the higher number of ETCs is just (over-)compensated by their weaker intensity (which would result in an equal or even weaker poleward heat transport)? Most of the additional cyclones in the AA experiment are the weak cyclones which occur at high latitudes (Figures 9 and 11 in the original manuscript) and probably contribute very little to the large-scale energy transport. This was briefly discussed in lines 409-412 in the original manuscript. These sentence have been revised slightly.

7. L236-237: What about the latitude distribution during maximum intensity? Is there also an equatorward shift in AA, as for genesis and lysis? Does this shift result from the equatorward shift of the baroclinic zone in this experiment? And how does this response compare to polar amplification scenarios of full GCMs? (see also my second major comment) The median values of the latitude of maximum vorticity are shown in Table 2 of the original manuscript. These values show that there is an equatorward shift in AA of 1.1 degrees compared to the control. We have added a sentence to the manuscript stating that the response of the latitude of maximum vorticity is the same as the response of the genesis and lysis latitudes. This is consistent with the equatorward shift in the jet which is now shown in the newly added figure to the manuscript and is consistent with the small decrease and equatorward shift in the low level temperature gradient shown in Figure S1 in the revised supplementary material.

8. L258: The last part of this long sentence ("ETCs with the same intensity...") is not clear to me – can you rephrase? This has now been revised.

9. L274: What does the "s" mean in "(mm / 6 hr) s"? These units are for the slope (gradient) of the best fit line. Since the gradient has the same units as precipitation divided by vorticity the $s$ here refers to seconds. Since this is standard SI units, we do not add an explanation to the manuscript.

10. L277-279: "Thus, we can conclude that there is a stronger relationship between maximum vorticity and precipitation in the SST4 experiment than in both the control or the AA experiment and also that the AA experiment has a stronger relation that the control." I think this statement is misleading, because a larger slope does not imply a stronger linear relationship, right? What determines the strength of the linear relationship is the r-value, which is actually smallest in SST4, largest in AA, and in between in Control. Hence, strictly speaking, your sentence above is not correct. However, as you explain in the subsequent sentence, it is about how much precipitation changes with a given vorticity change. So I would just rephrase the sentence above somehow (plus at a few other locations in the manuscript, where you write about the "strength of the relationship" referring to the slope). Yes we agree here in that "stronger relationship" is the incorrect wording as it does imply that the r-value is larger - which is not the case. We have revised this (and similar statements elsewhere) to be "Thus, we can conclude that there is a greater dependency between precipitation and maximum vorticity in the SST4 experiment than in...."

11. L279-282: "This means that for the same increase in maximum vorticity, precipitation increases more in the SST4 and AA experiments compared to the control. Since the SST4 and AA experiments are both warmer than the control experiment this is to be expected based on the Clausius-Clapeyron relationship between temperature and vapour pressure." Is this really that obvious / to be expected? If only the C-C relationship would matter, then also the weak cyclones (low relative vorticity) would need to have equally enhanced precipitation, which would imply that the slope should stay the same, but the intersect would change, right? So, maybe one could also interpret the increased slope such that the stronger cyclones are better able to convert the enhanced moisture content into enhanced precipitation than the weaker cyclones (likely due to the stronger dynamically induced forcing). Does that make sense? We agree with the reviewer in that it is very unlikely to be as simple as we stated it. We agree that if only the CC relationship mattered, precipitation would increase by the same percentage for all ETCs regardless of their intensity so yes the slope would stay the same but the intercept would increase. It is a valid hypothesis that stronger cyclones, with stronger forcing from e.g. thermal or vorticity advection, are better able to convert the extra moisture into precipitation - potentially as their ascent exceeds a threshold - however it it hard to test / prove this. We have revised this sentence and other parts of section 5 which include explanations for changes in the slope.

12. Fig. 4 (and related figures): Could you change the color of the SLP contours? It's very hard to distinguish them from the black TH@850 contours. We have re-made figure 4 to make the contours easier to see. Specifically, we have removed the range rings and have moved the 850-hPa potential temperature contours to the new figure showing the low-level potential vorticity.

13. L288: Did you consider just showing equivalent potential temperature instead of potential temperature and total column water? It might be the better variable to show temperature distribution, the sharpness of the fronts, and the moisture availability at once. But it's a matter of taste, I guess. We agree that this is a matter of taste and we prefer to keep the moisture and temperature aspects of the fronts separate here. Also both potential temperature and total column water can be directly output from the OpenIFS simulation whereas we would need to compute equivalent potential temperature off-line.

14. L293: It's interesting to see these four clusters. Do you know about similar clusters in reanalysis? If yes, do they look similar and could you refer to these studies? We also find these 4 distinct clusters very interesting and are not aware of similar cyclone structures (in terms of precipitation) having been identified in reanalysis. It would certainly be very interesting to perform this analysis in a future study. However, we hypothesis that more complex / more numerous ETC clusters may be found in reality due to geographical differences, mountain ranges, storm tracks that are not purely zonal etc. We should note that there are previous studies which have used similar clustering techniques to identify different ETC structures. For example, Catto (2018) used k-means clustering with the 250-hPa wind speed and anomalies from the zonal mean potential temperature on the tropopause as input to separate ETCs in the south Pacific. However, as far as we are aware no previous studies have attempted to use clustering techniques to separate cyclones based on their precipitation patterns (i.e with precipitation as the input to the cluster algorithm).

15. Fig. 6: I very much like to see the upper-level structure of the different clusters. Just for my understanding, could it be that some jet maxima we see in this figure are related to the subtropical jet? This also relates a bit to my first major comment regarding how the background state looks like on average. The upper level structure can be seen in the potential vorticity on the 315 K surface which is shown in Figure 6 of the original manuscript. These wind speeds are at 300 hPa so are likely at least partly related to the subtropical jet. The new figure of the zonal mean zonal winds that has been added to the revised manuscript shows that the core of the jet is at approximately 180 hPa in the control simulation so 300-hPa is below this. However, the changes in the zonal wind are strongly evident at the 300 hPa level.

16. L305-306: You might also add here that the upper-level PV values downstream are lowest in this cluster, probably because of the strongest diabatic erosion of PV downstream due to the WCB ascent. We have added a sentence here to highlight the low PV values downstream.

17. L348-349: Do you know why the cold front ETCs (in the control) are more south than the warm front ETCs? Is this also the case when comparing climatologies of Norwegian cyclones with Shapiro-Keyser cyclones? I assume thanks to this fact, the cold front ETCs can draw more from the subtropical moisture reservoir and thus become more intense with more precipitation than the warm front ETCs, right?

Firstly, yes, the fact that the cold front ETCs are further south enables them to access more moisture and ultimately produce more precipitation than the warm front ETCs. Unfortunately, we do not have a explanation for why the cold front ETCs reach their maximum intensity more equatorward than the warm front ETCs in the control simulation. As far as we are aware there are no published climatologies showing the spatial distribution of Norwegian-type cyclones and Shapiro-Keyser type cyclones, but previous studies do show (1) Shapiro-Keyser cyclones develop

in confluent flow as is typically found in the jet entrance region and Norwegian cyclones in diffluent flow in the jet exit regions (Schultz and Zhang (2007)) and (2) Cyclones with their centre north of the environmental temperature gradient tend to experience stronger cold frontogenesis to the south-west of the cyclone centre and cyclones south of the temperature gradient maximum have strong warm frontogenesis to the north-east (Keyser *et al.* (1988), Davies (1999)). Both of these findings indicate that cyclones with strong warm fronts potentially occur farther south than those with strong cold fronts which is opposite to what we observed in our aqua-planet simulation.

18. L352: I think there is something off with the sentence "The mean latitude that ETCs in the centre ETC cluster reach their maximum vorticity..." We have revised this sentence.

19. L367-368: Regarding the fact that the tropopause level might change in the different simulations (see also my first major comment), did you also look at other vertical levels when investigating upper-level PV? The tropopause level does increase in the SST4 simulation compared to the control simulation - see the new Figure 2 in the revised manuscript. However, we did not consider any other isentropic surfaces in our analysis.

20. L397: Which one is the "small cluster" you mention here? Sorry this should be the centre ETC - we changed the name of this cluster at some point and this instance was not updated. It is now fixed.

21. L415-419: I find the strongest correlation between vorticity and precipitation in the warm front cluster interesting, but I think I don't understand the reason. I would have expected it to be stronger in the cold front cluster. Do you have an explanation for this? This is an interesting result - that both the slope and the Pearson's correlation coefficient are largest for the warm front ETC in all experiments. We have two hypotheses: (1) that precipitation is less variable for the same maximum vorticity (and hence the large r) is because there is less convective precipitation and (2) that the position of the precipitation on the warm front induces a low-level PV anomaly which extend downstream of the ETC centre, rather than being just co-located with the ETC centre as is the case in the other clusters, means that there may be a stronger coupling between the precipitation and maximum vorticity. We have revised parts of section 7 and the conclusions in an attempt to explain this result.

22. L507-510: "The cold front ETC sees an increase in its maximum vorticity with uniform warming whereas both the warm front and cyclone centre ETCs show no change in the maximum vorticity and the maximum vorticity of the weak ETC decreases." Isn't this somehow contradictory to the aforementioned fact that the vorticity-precipitation relationship is strongest and with the largest slope in the warm front ETCs, which get much more precipitation in the warm experiments? Or does this just show that we cannot understand everything from just looking at these two variables (see also my first major comment)? The previous statement was about comparing ETCs within an experiment; when this is done the warm front ETC has the largest slope and correlation coefficient. This current statement is about comparing the same cluster in different experiments. The warm front ETC is SST4 has a larger slope (0.267 versus 0.225) which is consistent with the no change in vorticity and and increase in precipitation in the warm front cluster between SST4 and the control simulation.

**Typos / suggestions for rephrasing**

1. L32: Replace "Even" with "Already"? Changed.

2. L33: Comma after "ascending" Comma added.

3. L40: Replace the second "using" with "based on" Changed

4. L70: Rephrase to "that the number of ETCs associated with extreme precipitation may triple" Revised as suggested.

5. L72: Comma or semicolon after "simulation" comma added.

6. L95: Delete "will be on" Deleted

7. L108: "setup" rather than "set up"? Change to setup.

8. L126: Write consistently either "aquaplanet" or "aqua-planet". Now revised to be aqua-planet everywhere.

9. L141: "those previous simulations" corrected.

10. L167: "due east" $->$ "to the east"? revised as suggested.

11. L238: Comma after "Lastly" comma added.

12. L265: Rephrase to "The same as Figure 3 but for precipitation 24 and 0 hours before the time of maximum vorticity are shown in the Supplementary material." This is now revised.

13. L279: "... stronger relation than the control." Correct that to than and have also added a comma to this sentence to make it clearer.

14. L286: "... than in other simulations." corrected that to than.

15. L314-315: Delete "associated with them" deleted

16. L327: Delete "associated with it" deleted

17. L342: "whereas the weak ETCs have very different shaped distributions" revised.

18. L354: Delete "associated with them" deleted

19. L389: Delete "associated with then" We did not revised this as we think it is appropriate the say precipitation is associated with an ETC. In contrast, an ETC has a maximum vorticity.

20. L396: ")" after "Fig. 10" added.

21. L397: "with the weak ETC cluster" Added "the"

22. L425: Delete "associated with them" We did not modify this - we think it is appropriate to say precipitation is associated with a ETC.

23. L493: Typo in "vorticity corrected.

**References**

Catto, J. L. (2018). A new method to objectively classify extratropical cyclones for climate studies: Testing in the southwest pacific region. *J. Climate*, **31**(12), 4683–4704.

Chen, G., Plumb, R. A., and Lu, J. (2010). Sensitivities of zonal mean atmospheric circulation to sst warming in an aqua-planet model. *Geophys. Res. Lett.*, **37**(12).

Davies, H. C. (1999). Theories of frontogenesis. *The Life Cycles of Extratropical Cyclones*, pages 215–238.

Hartmann, D. L. (2015). *Global Physical Climatology*, volume 103. Elsevier.

Keyser, D., Reeder, M. J., and Reed, R. J. (1988). A generalization of petterssen's frontogenesis function and its relation to the forcing of vertical motion. *Mon. Wea. Rev.*, **116**(3), 762–781.

Möbis, B. and Stevens, B. (2012). Factors controlling the position of the intertropical convergence zone on an aquaplanet. *Journal of Advances in Modeling Earth Systems*, **4**(4).

Neale, R. B. and Hoskins, B. J. (2000). A standard test for AGCMs including their physical parametrizations: I: The proposal. *Atmospheric Science Letters*, **1**(2), 101–107.

Priestley, M. D. and Catto, J. L. (2022). Future changes in the extratropical storm tracks and cyclone intensity, wind speed, and structure. *Weather Clim. Dynam.*, **3**, 337–360.

Schultz, D. M. and Zhang, F. (2007). Baroclinic development within zonally-varying flows. *Q. J. Roy. Meteor. Soc.*, **133**(626), 1101–1112.

Sinclair, V. A., Rantanen, M., Haapanala, P., Räisänen, J., and Järvinen, H. (2020). The characteristics and structure of extra-tropical cyclones in a warmer climate. *Weather Clim. Dynam.*, **1**(1), 1–25.

---

## Referee Report (RR1)

**Second review of "The relationship between extra-tropical cyclone intensity and precipitation in idealised current and future climates" by Sinclair and Catto re-submitted to Weather and Climate Dynamics**

**General comments**

I thank you for considering and carefully replying to all the reviewers' comments. I think the (very interesting!) additional analysis of the changes in the mean atmospheric state now greatly "round off" the manuscript and make many results regarding the linear relationship between vorticity and precipitation better understandable/interpretable. Therefore, I suggest to accept the revised manuscript after considering the following few very minor comments that all refer to how things are phrased (the line numbers below refer to the line numbers in your tracked-changes document). Note that particularly in the new text passages there are still quite a few typos, of which some I mention below.

**Minor comments**

L5-8: I think it's great that you now discuss this aspect (i.e. similar vs. changing slope, and the potential role of changes in the background state) so clearly throughout the manuscript, but I wonder whether this "hypothesis" sentence is really at the right location here in the abstract? Could you mention this hypothesis later, after the finding that the slopes slightly change and what this might mean (see next comment)? I'm not sure what is better, but in the current form I find the sentence a bit isolated, because later you don't really refer back to this hypothesis...

L13-14: This sounds very technical now for an abstract ("slope of the linear regression line is statistically larger"), and, moreover, not unambiguous, as it's not clear which slope you actually refer to. Why not writing something like: "The amount of precipitation for ETCs with a specific vorticity is higher in the uniform warming and polar amplification simulations than in the control simulation (i.e., the slope of the linear regression between vorticity and precipitation is larger), indicating that changes…". Furthermore, could you add the explanation for this conclusion, i.e. the hypothesis mentioned earlier, right here instead (see previous comment), and then mention the potentially additional processes that might compete with diabatic heating (i.e., reduction in baroclinicity, which you currently don't mention at all in the abstract)? I think this would make it easier to follow the line of argumentation what a change in slope could mean. I hope you understand my suggestion…

L19: You both use "dependence" and "dependency" in the abstract, so I would consistently use only one.

L21: Typo "voricity"

L26: Comma before "whereas"

Figure 2 captions: Maybe write "difference between SST4 and control" (i.e., not just in brackets)

L256: Change to "and causes a decrease in the Eady…"

L258-259: Maybe change to "rather than a decrease in the meridional temperature gradient, which barely changes"

L259-261: I think you should help the reader to see where (and if) the lifting of the tropopause can be seen in these figures. I assume you could kind of see it in the changes in the Brunt Väisälä frequency, right? But how exactly? Is the tropopause basically going up by the vertical extent of the negative Brunt Väisälä frequency anomaly? At least, this lift is not obvious to me at first glance...

L261: "the jet to move equatorwards"

L264: "related to a decrease"

L321: "largest slopes and correlation coefficients occur"; furthermore, I guess you refer to Figure 4 when you write "The same as Figure 3, but…"?

L345 and L347: Change back to "feedback" as it's a noun there

L346-349: It's nice that you now include this sentence here, but it is a bit "heavy" to read and not very specific. In fact I liked the wording you used in your response document more, when you said something like "the increasing slope might still indicate an increased diabatic feedback on vorticity, which, however, might be masked by the counteracting reduction in Eady growth rate" -> I think the use of "mask" or something similar might be helpful here…

Figures 5 and 6: Did you leave the range rings here on purpose, although you removed them in Fig. 4? I think you could just remove them everywhere.

L495: "in the number of weak ETCs"

L509: "compared to cold front ETCs"

L542: Something is off with "… as with uniform uniform these ETCs are less…"

L555: "means that precipitation"

L557-558: "do not act to intensify warm front ETCs"

L560-562: I don't understand this sentence here, particularly the second part of the sentence "…, the weak ETCs in this cluster see…"

Conclusion: Make sure you stay with one tense, because you start with past tense but then, at least partly, fall into present tense (for instance at L577).

L578: "uniform warming"

L595: "feedback" instead of "feed back" as it's a noun here, right?

---

## Author Response (AR2)

**Response to reviewers - The relationship between extra-tropical cyclone intensity and precipitation in idealised current and future climates**

Victoria Sinclair and Jennifer Catto

May 6, 2023

**1   Editor's comments**

Thanks for the detailed responses and comprehensive revisions of your manuscript. Your paper has been assessed once again by the two reviewers and both recommend publication in WCD. One of the reviewers has a few minor suggestions that I'd ask you to take into account when preparing the final version. In addition, please also consider the following very minor points:

1. Line 46 ("This diabatic heating..."): I think it would be nice to support this sentence with a reference, potentially to one of the earlier papers on this linkage, such as Stoelinga (1996). We have added a reference to Stoelinga (1996)

2. Line 67: Held and Soden didn't really look at extremes; maybe add another reference, such as Allen and Ingram (2002). We have changed this reference as suggested

3. Line 76 ("large ensemble of climate model simulations"): It would be more precise to call this a "single model initial condition large ensemble", just to make clear that it is not a CMIP-type analysis. This is now revised to read "analysed a 30-member initial condition climate model ensemble"

4. Line 135: "extrapolated" instead of "interpolated"? Yes, this is more accurate and we have changed it.

5. Line 166: Is there any reason for only looking at the Northern Hemisphere? Taking into account both hemispheres would have been a cheap way to increase your sample size. No, there is no major reason for only considering one hemisphere. One very small reason was that one of the metrics we used to determine if the simulations had spun up into balance was the difference in the total precipitation between each hemisphere (which should be close to zero). Furthermore, we felt our sample size of >10,000 cyclones in each experiment was large enough and it was just computationally simpler to run the simulations for 10 years rather than just 5 and consider two hemispheres.

6. Line 250: I'd find it helpful to specify that you're referring to the "horizontal" temperature gradient. We now write the "meridional temperature gradient" here.

7. Line 320, Figure 4: Are the units really correct, or is there a factor of $10^5$ missing?. Yes, you are correct. We have corrected the units here and elsewhere throughout the manuscript.

8. Line 528: "significant" instead of "significantly different". Revised as suggested.

9. Fig 3: Consider to use different symbols or line styles to make the figure more accessible for color-blind readers. We have updated Figure 3 so that each experiment has a different line marker in addition to be a different colour.

**Reviewer 2**

**General comments**

I thank you for considering and carefully replying to all the reviewers' comments. I think the (very interesting!) additional analysis of the changes in the mean atmospheric state now greatly "round off" the manuscript and make many results regarding the linear relationship between vorticity and precipitation better understandable/interpretable. Therefore, I suggest to accept the revised manuscript after considering the following few very minor comments that all refer to how things are phrased (the line numbers below refer to the line numbers in your tracked-changes document). Note that particularly in the new text passages there are still quite a few typos, of which some I mention below.

**Minor comments**

1. L5-8: I think it's great that you now discuss this aspect (i.e. similar vs. changing slope, and the potential role of changes in the background state) so clearly throughout the manuscript, but I wonder whether this "hypothesis" sentence is really at the right location here in the abstract? Could you mention this hypothesis later, after the finding that the slopes slightly change and what this might mean (see next comment)? I'm not sure what is better, but in the current form I find the sentence a bit isolated, because later you don't really refer back to this hypothesis... We have now moved this hypothesis sentence to the start of section 5 and also refer to it later on in section 5.

2. L13-14: This sounds very technical now for an abstract ("slope of the linear regression line is statistically larger"), and, moreover, not unambiguous, as it's not clear which slope you actually refer to. Why not writing something like: "The amount of precipitation for ETCs with a specific vorticity is higher in the uniform warming and polar amplification simulations than in the control simulation (i.e., the slope of the linear regression between vorticity and precipitation is larger), indicating that changes...". Furthermore, could you add the explanation for this conclusion, i.e. the hypothesis mentioned earlier, right here instead (see previous comment), and then mention the potentially additional processes that might compete with diabatic heating (i.e., reduction in baroclinicity, which you currently don't mention at all in the abstract)? I think this would make it easier to follow the line of argumentation what a change in slope could mean. I hope you understand my suggestion... We have edited this part of the abstract to include the mention of competing processes, and to make it less technical sounding, as suggested. However, the suggested line "The amount of precipitation for ETCs with a specific vorticity is higher in the

uniform warming and polar amplification simulations than in the control simulation" is not quite what we meant here, since this does not necessarily mean that the slope will be larger.

3. L19: You both use "dependence" and "dependency" in the abstract, so I would consistently use only one. We have revised this to only use "dependency"

4. L21: Typo "voricity" Corrected

5. L26: Comma before "whereas" Added

6. Figure 2 captions: Maybe write "difference between SST4 and control" (i.e., not just in brackets) We already write something very similiar: *"Shading shows the difference in the zonal wind speed between the control and SST4 simulation"* and therefore we have not made any revisions here.

7. L256: Change to "and causes a decrease in the Eady..." Sorry this was a typo and is now revised as suggested.

8. L258-259: Maybe change to "rather than a decrease in the meridional temperature gradient, which barely changes". Revised as suggested.

9. L259-261: I think you should help the reader to see where (and if) the lifting of the tropopause can be seen in these figures. I assume you could kind of see it in the changes in the Brunt Väisälä frequency, right? But how exactly? Is the tropopause basically going up by the vertical extent of the negative Brunt Väisälä frequency anomaly? At least, this lift is not obvious to me at first glance...We have now indicated this is the previous line when mentioning the lifting of the jet.

10. L261: "the jet to move equatorwards" Revised as suggested.

11. L264: "related to a decrease" Revised as suggested.

12. L321: "largest slopes and correlation coefficients occur"; furthermore, I guess you refer to Figure 4 when you write "The same as Figure 3, but..."? Occurs has now been changed to occur and the reference to the figure has been corrected.

13. L345 and L347: Change back to "feedback" as it's a noun there. Revised as suggested.

14. L346-349: It's nice that you now include this sentence here, but it is a bit "heavy" to read and not very specific. In fact I liked the wording you used in your response document more, when you said something like "the increasing slope might still indicate an increased diabatic feedback on vorticity, which, however, might be masked by the counteracting reduction in Eady growth rate" $->$ I think the use of "mask" or something similar might be helpful here...Thank you for the suggestions. We have now rephrased this sentence to make is clearer and easier to read.

15. Figures 5 and 6: Did you leave the range rings here on purpose, although you removed them in Fig. 4? I think you could just remove them everywhere. We think it is good to give an idea of the spatial scale on these figures and since Figures 5 and 6 are not too crowded we decided to keep them here.

16. L495: "in the number of weak ETCs" Corrected

17. L509: "compared to cold front ETCs" We have left this as is as we think it is correct.

18. L542: Something is off with "... as with uniform uniform these ETCs are less..." This has been revised to read "as with unform warming"

19. L555: "means that precipitation" Corrected

20. L557-558: "do not act to intensify warm front ETCs" Corrected

21. L560-562: I don't understand this sentence here, particularly the second part of the sentence "..., the weak ETCs in this cluster see..." What we meant here is that if you compare the best fit linear regression lines shown in Figure 14a and 14c, the left side (small vorticity values) increases (moves to higher precipitation values) more than the right hand side - although this is quite subtle and hard to see from the figures. It can also be computed using the slope and intercepts of both best fit lines. For the cold front cluster in the control simulation the equation of the best fit line is $TP = 0.176Vo + 2.391$ whereas in the AA experiment the best fit lines is $TP = 0.172Vo + 2.614$. These equations show that for a ETC with a maximum vorticity of 1 $\times 10^{-5}\text{s}^{-1}$ the 6 hr precipitation in the AA experiment is 0.215mm larger than in the control simulation. In contrast, for an ETC with a maximum vorticity of $1 \times 10^{-5}\text{s}^{-1}$ the corresponding value is 0.162 mm.

22. Conclusion: Make sure you stay with one tense, because you start with past tense but then, at least partly, fall into present tense (for instance at L577). We have revised the conclusions section to ensure that we use the past tense throughout.

23. L578: "uniform warming" Corrected

24. L595: "feedback" instead of "feed back" as it's a noun here, right? Yes, this is a noun so changed to feedback

**References**